# Mutant SF3B1 promotes malignancy in PDAC

Patrik Simmler[1,2], Eleonora I Ioannidi[2], Tamara Mengis[2†], Kim Fabiano Marquart[1,2], Simran Asawa[1], Kjong Van-Lehmann[3,4], Andre Kahles[3,4], Tinu Thomas[3,4], Cornelia Schwerdel[2], Nicola Aceto[1], Gunnar Rätsch[3,4,5,6], Markus Stoffel[1], Gerald Schwank[2]*

[1]Department of Biology, Institute of Molecular Health Sciences, ETH Zurich, Zurich, Switzerland; [2]Institute of Pharmacology and Toxicology, University of Zurich, Zurich, Switzerland; [3]Department of Computer Science, Biomedical Informatics Group, ETH Zurich, Zurich, Switzerland; [4]Swiss Institute of Bioinformatics, Lausanne, Switzerland; [5]Department of Biology, ETH Zurich, Zurich, Switzerland; [6]Biomedical Informatics Research, University Hospital Zurich, Zurich, Switzerland

*For correspondence:
schwank@pharma.uzh.ch

Present address: [†]Center of Experimental Rheumatology, University Hospital Zurich, Zurich, Switzerland

Competing interest: The authors declare that no competing interests exist.

**Abstract** The splicing factor SF3B1 is recurrently mutated in various tumors, including pancreatic ductal adenocarcinoma (PDAC). The impact of the hotspot mutation SF3B1[K700E] on the PDAC pathogenesis, however, remains elusive. Here, we demonstrate that Sf3b1[K700E] alone is insufficient to induce malignant transformation of the murine pancreas, but that it increases aggressiveness of PDAC if it co-occurs with mutated KRAS and p53. We further show that Sf3b1[K700E] already plays a role during early stages of pancreatic tumor progression and reduces the expression of TGF-β1-responsive epithelial–mesenchymal transition (EMT) genes. Moreover, we found that SF3B1[K700E] confers resistance to TGF-β1-induced cell death in pancreatic organoids and cell lines, partly mediated through aberrant splicing of *Map3k7*. Overall, our findings demonstrate that SF3B1[K700E] acts as an oncogenic driver in PDAC, and suggest that it promotes the progression of early stage tumors by impeding the cellular response to tumor suppressive effects of TGF-β.

## Editor's evaluation

This important study investigates the oncogenic and disease promoting potential of the K700E mutation in the splicing factor SF3B1 in a mouse model for pancreatic ductal adenocarcinoma (PDAC), finding that this mutation can promote disease progression both in the presence or absence of Trp53. They further identify SF3B1K700E-induced missplicing of Map3k7 as a critical mechanism that enables isolated pancreas cancer cells to survive TGFβ-induced EMT, senescence and cell death. Together with experiments based on convincing methods that suggest a conserved function in human PDAC, and indicating a specific role for the SF3B1 K700E mutation in early stage tumors, this study makes a valuable contribution to understanding of mechanisms regulating PDAC transformation, and will be interesting to researchers investigating pancreatic cancer.

## Introduction

Genes involved in RNA splicing are frequently mutated in various cancer types (*Yoshida et al., 2011*). The splicing factor subunit 3b 1 (SF3B1) is amongst the most commonly mutated components of the splicing machinery, with high incidence in myelodysplastic syndromes (MDS; *Je et al., 2013*) and chronic lymphocytic leukemia (CLL; *Miao et al., 2019*). However, also in various solid tumors, SF3B1 is recurrently mutated, including uveal melanoma (UVM; *Furney et al., 2013*), breast cancer (BRCA; *Fu*

*et al., 2017*; *Maguire et al., 2015*; *Sun et al., 2020*), prolactinomas (*Li et al., 2020*), hepatocellular carcinoma (HCC; *Zhao et al., 2021*), and pancreatic adenocarcinoma (PDAC; *Bailey et al., 2016*; *Yang et al., 2021*). As part of the U2 small nuclear ribonucleoprotein (U2 snRNP) SF3B1 exerts an essential function in RNA splicing by recognizing the branchpoint sequence (BPS) of nascent RNA transcripts (*Wahl et al., 2009*; *Zhang et al., 2020*). This process is crucial for the definition of the 3′ splice site (3′ ss) of the upstream exon-intron boundary, a prerequisite for the accurate removal of introns (*Wahl et al., 2009*). It is well understood that hotspot mutations in SF3B1 at HEAT repeats 5–9 allow the recognition of an alternative BPS, resulting in the inclusion of a short intronic region into the mature messenger RNA (mRNA; *Alsafadi et al., 2016*; *Canbezdi et al., 2021*; *Darman et al., 2015*; *DeBoever et al., 2015*; *Kesarwani et al., 2017*). These alternatively spliced transcripts are prone to degradation through nonsense-mediated RNA decay (NMD) (*Darman et al., 2015*). Several recent studies have evaluated the mechanistic contribution of genes misspliced by oncogenic SF3B1 to tumor progression. So far, missplicing of *PPP2R5A* was found to increase malignancy through stabilizing c-Myc (*Liu et al., 2020a*; *Yang et al., 2021*), and aberrant *MAP3K7* splicing was reported to promote NF-κB-driven tumorigenesis (*Liu et al., 2021*).

Despite compelling evidence on the oncogenic role of mutated SF3B1 in hematologic malignancies, its contribution to the formation or progression of solid tumors is less understood. Since splicing deregulation has been reported as a hallmark of PDAC, with SF3B1 being recurrently mutated (*Bailey et al., 2016*), we aimed at elucidating the impact of the frequently occurring SF3B1$^{K700E}$ mutation to the pathogenesis of this tumor type. We demonstrate that *Sf3b1$^{K700E}$* increases malignancy in a mouse model for PDAC and decreases the sensitivity to TGF-β-responsive EMT genes. Experiments in pancreatic organoids and cell lines further provide evidence that SF3B1$^{K700E}$ protects from TGF-β-induced cell death and that TGF-β-resistance is partly mediated through missplicing of *Map3k7*. Together, our work suggests that SF3B1$^{K700E}$ exerts its oncogenic role in PDAC by dampening the tumor-suppressive effect of TGF-β.

## Results

### SF3B1$^{K700E}$ is a tumor driver during early stages of PDAC formation

RNA processing has been previously identified as a hallmark of pancreatic cancer in the published Pancreatic Cancer Australian (PACA-AU) cohort (*Bailey et al., 2016*). Validating these findings, we found that genes encoding for the splice factors RBM10, SF3B1, and U2AF1 are also frequently mutated in the Pancreatic Cancer Canadian (PACA-CA) cohort (*Figure 1—figure supplement 1A*). In accordance with its described function as a tumor suppressor (*Hernández et al., 2016*), 56% of the mutations found in RBM10 lead to a truncated protein. Conversely, in SF3B1 and U2AF1 the majority of mutations were missense mutations that occurred at hotspot sites, indicating a neomorphic function of the mutated proteins. Like in other cancer types, also in PDAC the most frequently found mutation in SF3B1 led to a lysine (K) to glutamic acid (E) change at position 700 (SF3B1$^{K700E}$; *Figure 1—figure supplement 1B*). Therefore, we experimentally tested if the SF3B1$^{K700E}$ mutation contributes to PDAC malignancy by generating a mouse model where the *Sf3b1$^{K700E}$* mutation is specifically activated in the pancreas using Ptf1a-Cre (*Figure 1A*, *Figure 1—figure supplement 1C*). First, we tested if *Sf3b1$^{K700E/+}$* alone is sufficient to induce PDAC formation. However, heterozygous activation of the *Sf3b1$^{K700E}$* allele did not have any effect on survival of mice or weight of the pancreas after 300 days (*Figure 1B and C*). Furthermore, assessing the pancreata histologically for two prognostic markers for PDAC, Cytokeratine-19 (CK19) and Mucin 5AC (MUC5AC), revealed no difference to wild type control mice (CK19 was restricted to pancreatic ducts and MUC5AC was not expressed; *Figure 1—figure supplement 1D, E*). Next, we assessed if the K700E mutation increases aggressiveness of PDAC, and crossed the *Sf3b1$^{K700E}$* allele into the *Ptf1a-Cre; Kras$^{G12D/+}$; Trp53$^{fl/fl}$* (KPC) mouse model, which is known to induce PDAC within 2–3 months (*Bardeesy et al., 2006a*; *Hingorani et al., 2005*; *Marino et al., 2000*). Importantly, KPC-*Sf3b1$^{K700E/+}$* animals displayed a significantly shorter survival (mean survival 57 days vs. 64 days), and an increased tumor size compared to KPC mice at the age of 9 weeks (*Figure 1D–F*). However, since at this timepoint similar amounts of tissue fibrosis and CK19 positive cells were observed in *Sf3b1$^{K700E/+}$* mutant vs. *Sf3b1$^{+/+}$* KPC tumors (*Figure 1—figure supplement 1F–I*), we next assessed if SF3B1$^{K700E}$ already promotes malignancy at early stages of tumorigenesis. Indeed, when we analyzed pancreata of 5-week-old mice, already 67% of KPC-*Sf3b1$^{K700E/+}$* mice vs. 25% of KPC mice

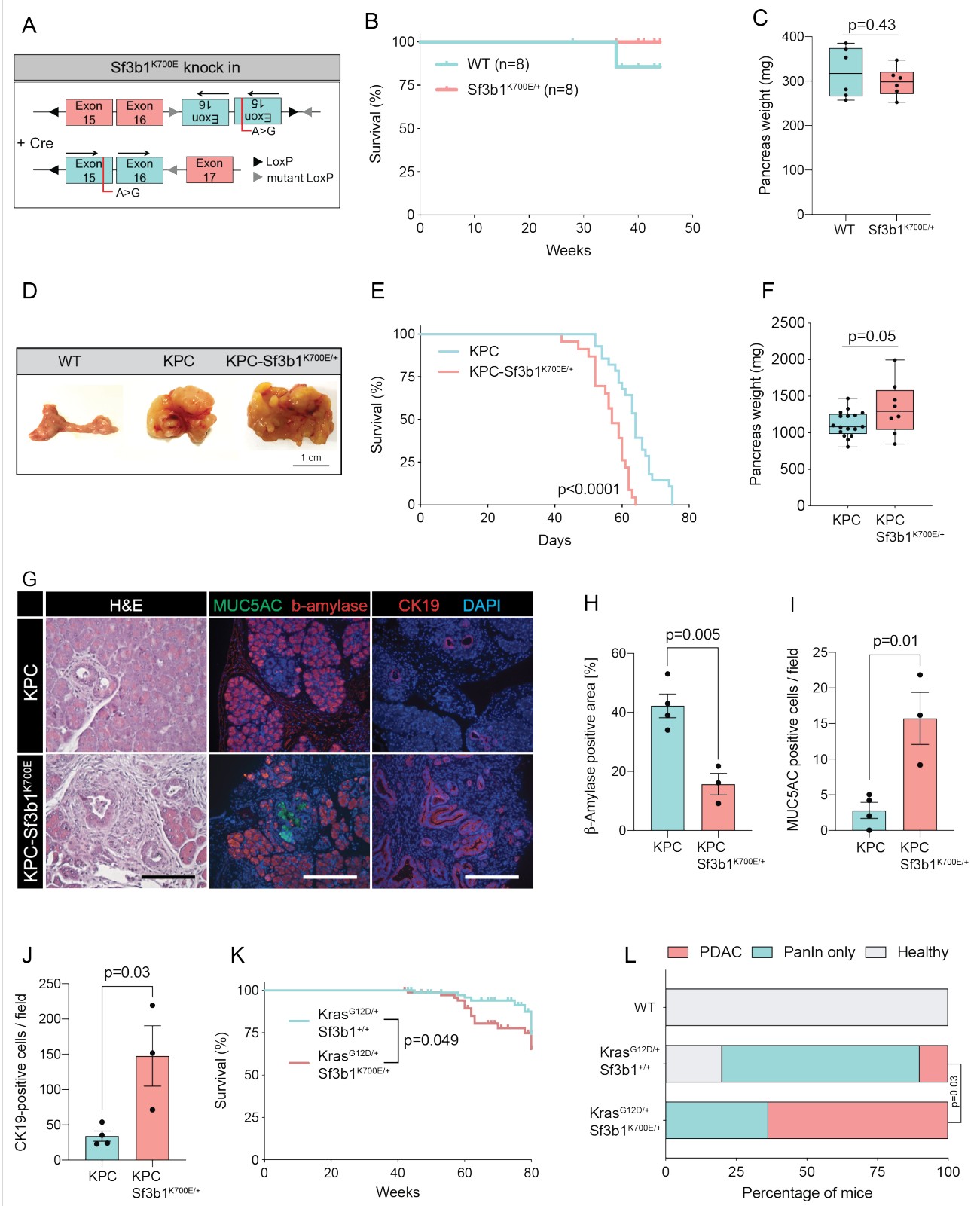

**Figure 1.** Sf3b1^K700E increases aggressiveness of murine PDAC. (**A**) Schematic overview of the *Sf3b1^K700E/+* knockin model. Arrows indicate direction of the translatable code of the gene. An A to G conversion at base position 2098, induced by Cre-recombination, results in a lysine to glutamic acid mutation at amino acid position 700 of the protein. (**B**) Survival of WT and *Sf3b1^K700E/+* mice followed over 300 days (n=8). (**C**) Pancreas weight of WT and *Sf3b1^K700E/+* mice at 300 days of age (3 males and 3 females for each genotype). Two-tailed unpaired t-test was used to compute the indicated

*Figure 1 continued on next page*

*Figure 1 continued*

p-value. (**D**) Representative photographs of WT, KPC, and KPC-Sf3b1$^{K700E/+}$ pancreata of mice 9 weeks old mice. (**E**) Survival of KPC and KPC-Sf3b1$^{K700E/+}$ mice, p-value was determined by Log-rank (Mantel-Cox) testing. (**F**) Pancreatic weight of KPC and KPC-Sf3b1$^{K700E/+}$ mice at 9 weeks of age. Two-tailed unpaired t-test was used to compute the indicated p-value. (**G**) Representative micrograph images of H&E, MUC5AC, ß-amylase and CK19 staining of KPC and KPC-Sf3b1$^{K700E/+}$ pancreata of 5-week-old mice. Scale bar is 50 μM (H&E) or 100 μM (IF). (**H**) Quantification of β-amylase (**H**), MUC5AC (**I**) and CK19 (**J**) staining shown in (**G**). Per specimen, the average value of 5 random microscopy fields is displayed, a two-tailed unpaired t-test was used to compute the indicated p-value. Error bar represents standard error of the mean (SEM). (**K**) Survival of *Kras*$^{G12D/+}$ and *Kras*$^{G12D/+}$; *Sf3b1*$^{K700E/+}$ mice. p-Value was determined by Log-rank (Mantel-Cox) testing. (**L**) Percentage of mice at 43 weeks of the indicated genotypes showing PanINs (blue) or PanINs and PDAC formation (red). p-Value indicates significance of the difference in PDAC formation, computed by Chi-square test.

The online version of this article includes the following figure supplement(s) for figure 1:

**Figure supplement 1.** Sf3b1$^{K700E}$ increases aggressiveness of murine PDAC.

developed PDAC. Moreover, we observed an increase in acinar-to-ductal metaplasia (ADM) formation and in the number of CK19 and MUC5AC positive cells, and a reduction in ß-amylase staining (*Figure 1G–J*, *Figure 1—figure supplement 1J, K*). Further supporting that SF3B1$^{K700E}$ already plays a role in early stages of carcinogenesis, *Sf3b1*$^{K700E/+}$ also had an effect on tumor malignancy when introduced into *Kras*$^{G12D/+}$ (KC) mice, which is a model for pre-cancerous pancreatic neoplasms with sporadic PDAC formation after a prolonged latency period (*Hingorani et al., 2005*). At 43 weeks of age, the area of neoplastic pancreas tissue was increased, and in 64% of KC-*Sf3b1*$^{K700E/+}$ mice vs. 10% of KC mice the lesions had already progressed to PDAC (*Figure 1L*; *Figure 1—figure supplement 1L*). In addition, the time of disease-free survival was significantly shortened in KC-*Sf3b1*$^{K700E/+}$ mice vs. KC mice (*Figure 1K*). Finally, in a model for advanced cancer, where KPC and KPC-*Sf3b1*$^{K700E}$ cells were harvested from fully developed PDAC tumors and expanded in vitro before being orthotopically transplanted, we did not observe differences in tumor growth (*Figure 1—figure supplement 1M, N*). Together, our data demonstrate that the SF3B1$^{K700E}$ contributes to PDAC malignancy by accelerating the formation of precursor lesions.

## SF3B1$^{K700E}$ reduces expression of EMT genes in pancreatic tumors

In order to elucidate the functional impact of SF3B1$^{K700E}$ on the transcriptome, we isolated cancer cells of mouse tumors by fluorescence-activated cell sorting (FACS) of Epithelial cell adhesion molecule (EpCAM) positive cells and performed RNA-sequencing (RNA-seq) (*Supplementary file 1*). High purity of isolated tumor cells was confirmed by the absence of sequencing reads for *Trp53* exons 2–10, which are excised via Cre-recombination specifically in tumor cells (*Figure 2—figure supplement 1A*), and by the presence of the *Sf3b1*$^{K700E}$ mutation in 38% of the transcripts (*Figure 1D*). Principal component analysis separated the sequenced replicates (3 KPC and 4 KPC-Sf3b1$^{K700E/+}$ tumors) according to the genotype, indicating a major impact of the K700E mutation on the transcriptome (*Figure 2—figure supplement 1B*).

We next performed gene set enrichment analysis (GSEA), which revealed IFN-α-response as the most significantly enriched pathway in KPC-Sf3b1$^{K700E/+}$ tumor cells (*Figure 2—figure supplement 1C*). This result is in line with a previous study, which found that aberrant splicing caused by SF3B1 inhibition or oncogenic SF3B1 mutations induces an IFN-α-response through retinoic acid-inducible gene I (RIG-I) mediated recognition of cytosolic aberrant RNA-species (*Chang et al., 2021*). Interestingly, epithelial-mesenchymal transition (EMT) emerged as the most significantly attenuated gene set in KPC-Sf3b1$^{K700E/+}$ cells (*Figure 2A and B*, *Figure 2—figure supplement 1C*). We first confirmed downregulation of the most significantly depleted gene of the EMT gene set, the glycoprotein Tenascin-C (*Tnc*), by qPCR on additional KPC-Sf3b1$^{K700E/+}$ tumor samples (*Figure 2C*) and by histology in KPC-Sf3b1$^{K700E/+}$ PDAC sections (*Figure 2D and E*). Next, we assessed if the reduction of EMT genes was induced cell-autonomously by the *Sf3b1*$^{K700E/+}$ mutation, or if it was an indirect consequence of the altered micro-environment in *Sf3b1*$^{K700E}$ KPC tumors. We therefore compared the expression of the 15 most significantly depleted EMT genes (*Figure 2B*) in vitro in *Sf3b1*$^{K700E}$ vs. *Sf3b1* WT KPC pancreatic organoids. Importantly, 71% of the analysed genes were significantly reduced in KPC-Sf3b1$^{K700E/+}$ vs. KPC organoids, with none of the genes showing a trend towards elevated expression (*Figure 2F*). To rule out that differences in EMT gene expression could be a consequence of differences in the tumor stage between *Sf3b1*$^{K700E}$ versus *Sf3b1* WT KPC tumors, we next established non-cancerous pancreatic organoids from LSL-*Kras*$^{G12D/+}$; *Trp53*$^{fl/fl}$; *Sf3b1*$^{flK700E/+}$ or *Sf3b1*$^{+/+}$ and LSL-*Kras*$^{G12D/+}$; *Trp53*$^{fl/}$

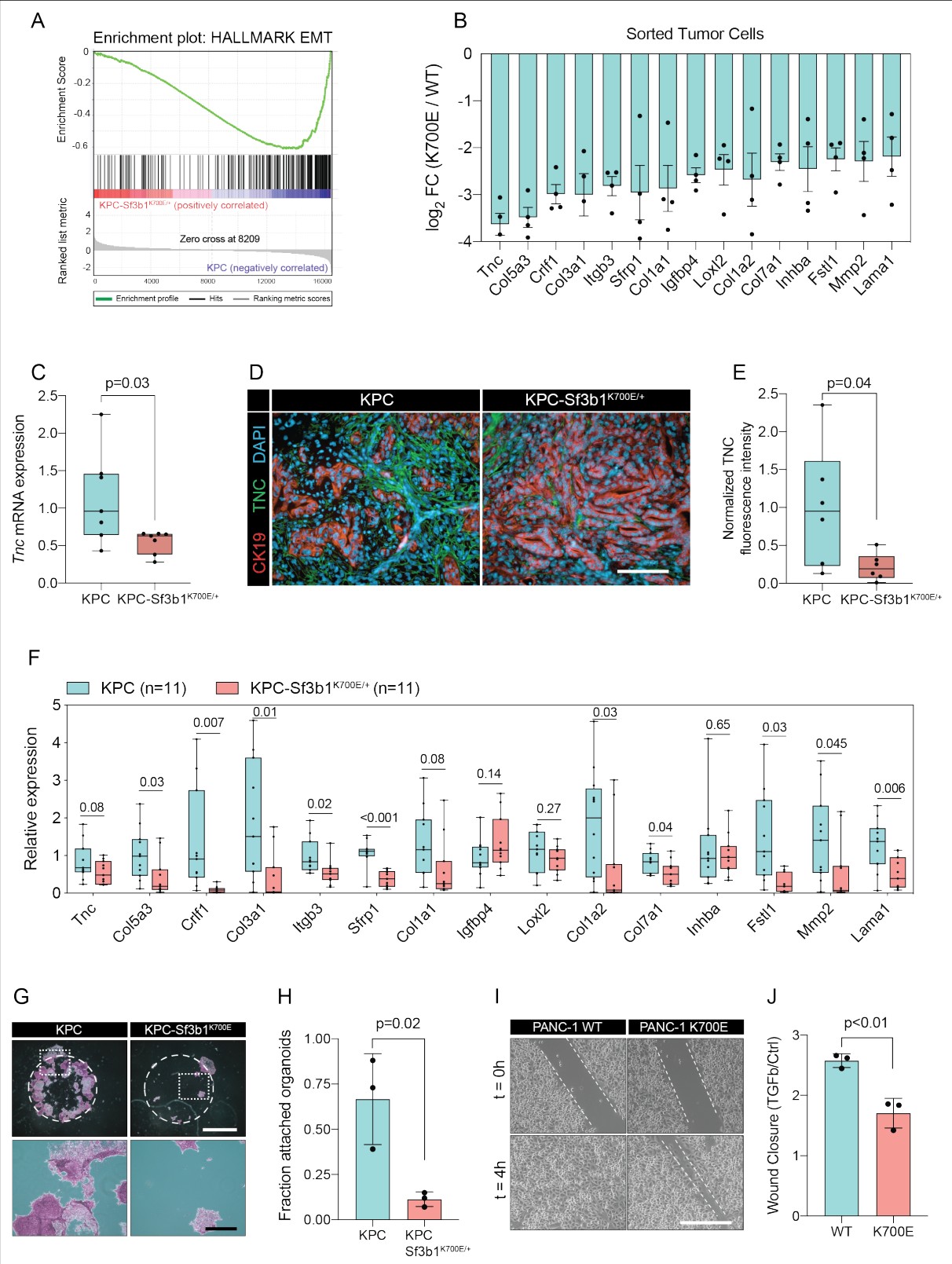

**Figure 2.** *Sf3b1^K700E* induces downregulation of EMT. (**A**) Gene-set enrichment analysis (GSEA) enrichment plot of epithelial-mesenchymal transition (EMT), representing the most deregulated pathway of the GSEA-Hallmark pathways when comparing KPC (n=3) and KPC-Sf3b1^K700E/+ (n=4) sorted tumor cells. (**B**) Top 15 of downregulated genes of the GSEA-EMT gene list in sorted KPC-Sf3b1^K700E/+ cells (FDR < 0.05, logCPM >1). (**C**) *Tnc* expression in KPC (n=7) and KPC-Sf3b1^K700E/+ (n=7) tumors, assessed by RT-qPCR. Error bar represent SEM. Two-tailed unpaired t-test was used to compute the

*Figure 2 continued*

indicated p-value. (**D**) Representative Immunofluorescence staining of CK19 (red) and TNC (green) in murine PDAC samples, counterstained with DAPI (blue). Scale bar is 50 μm. (**E**) Quantification of TNC staining in KPC (n=6) and KPC-Sf3b1$^{K700E/+}$ (n=6) tumors. The averaged area of TNC staining in three randomly chosen fields per tumor specimen was compared by a two-tailed unpaired t-test. (**F**) The expression of the EMT genes displayed in (**B**) was assessed by RT-qPCR in tumor cells (KPC, n=11 and KPC-Sf3b1$^{K700E/+}$ n=11) after one passage of ex-vivo culture. For analysis, Ct-values of the indicated genes were normalized to *Actb* and a two-tailed unpaired t-test was used to compute the indicated p-values. (**G**) Representative micrographs of KPC (n=3) and KPC-Sf3b1$^{K700E/+}$ (n=3) cancer organoid lines treated with TGF-β1 (10 ng/ml) for 48 hr. Matrigel was detached prior to staining with crystal violet, allowing quantification of cells migrated through the matrigel matrix and attached at the cell culture plate. Scale bar is 1 mm (panel above) or 100 μm (panel below). (**H**) Quantification of micrographs shown in (**G**). The fraction of attached organoids was calculated by dividing the number of attached organoids by the number of total organoids contained in the matrigel dome. The experiment was performed independently three times for every cell line, the average of all replicates is shown. Error bar represents SEM. Two-tailed unpaired t-test was used to compute the indicated p-value. (**I**) Representative micrographs of wound healing assay of PANC-1 WT and PANC-1 SF3B1$^{K700E/+}$ cells pre-treated with TGF-β1 (10 ng/ml) for 24 hours at the indicated time points after performing the scratch. Scale bar is 100 μm. (**J**) Quantification of wound healing assay of PANC-1 WT and PANC-1 SF3B1$^{K700E/+}$ cells, displaying the ratio of wound-closure of cells treated with TGF-β1 or without (Ctrl). The experiment was independently performed three times. Error bar represents standard deviation (SD). A two-tailed unpaired t-test was used to compute the indicated p-values.

The online version of this article includes the following figure supplement(s) for figure 2:

**Figure supplement 1.** *Sf3b1$^{K700E}$* induces downregulation of EMT.

---

$^{fl}$ mice and induced recombination in vitro through lentiviral Cre transduction. Importantly, also in this experimental setup 55% of the analysed EMT genes were significantly downregulated in *Sf3b1$^{K700E}$* vs. *Sf3b1* WT organoids, with only one of the analysed genes showing a minor trend for elevated expression (*Figure 2—figure supplement 1D*). Together, these data indicate that *Sf3b1$^{K700E}$* mediates downregulation of EMT-related genes in PDAC a cell autonomous manner.

## SF3B1$^{K700E}$ confers resistance to TGF-β1-induced apoptosis in organoids and cell lines

The two major EMT-promoting cytokines are TNF-α and TGF-β (*Bulle and Lim, 2020*). We therefore determined if the EMT genes that are most significantly downregulated by *Sf3b1$^{K700E}$* in KPC tumours are induced by TNF-α or TGF-β. Importantly, we found that 80% of the analysed genes were strongly induced by TGF-β in tumour derived KPC cells (*Figure 2—figure supplement 1E*), whereas TNF-α significantly upregulated only 20% of these genes (*Figure 2—figure supplement 1F*). Furthermore, we observed that in vitro induced *Sf3b1$^{K700E}$* KPC cells fail or only partially induce 6 of the 9 assessed EMT genes when stimulated with TGF-β (*Figure 2—figure supplement 1G*). In line with these results, in vitro induced *Sf3b1$^{K700E}$* KPC organoids exhibit a sixfold decrease in matrigel invasion compared to KPC control organoids when stimulated with TGF-β (*Figure 2G and H*). Moreover, human PANC-1 cells expressing SF3B1$^{K700E}$ display a reduced migratory rate compared to control PANC-1 cells expressing wildtype SF3B1 in a wound healing assay upon TGF-β stimulation (*Figure 2I and J*).

In pancreatic lesions TGF-β induces EMT, followed by apoptosis of the affected cells in a process termed lethal EMT (*David et al., 2016*). This prompted us to speculate that SF3B1$^{K700E}$ could drive PDAC progression by reducing sensitivity of epithelial cells to TGF-β-mediated apoptosis. Performing immunofluorescence staining for cleaved caspase 3 in KPC tumors, we first confirmed that the majority of apoptotic cells reside in the lumen of PanINs (*Figure 3—figure supplement 1A, B*), and that these cells are negative for the epithelial marker E-cadherin and positive for the mesenchymal marker Fibronectin-1 (*Figure 3—figure supplement 1C*; *Hruban et al., 2006*). We then analysed *Sf3b1$^{K700E}$* tumors by immunofluorescence staining. In line with our hypothesis, we observed a reduction in luminally extruded cells and a reduction in cleaved caspase 3-positive cells (CC3, *Figure 3A and B*, *Figure 3—figure supplement 1D*). To further analyse the impact of *Sf3b1$^{K700E}$* on the tumor suppressive effect of TGF-β, we exposed *Sf3b1* WT and *Sf3b1$^{K700E}$* KPC tumor organoids to TGF-β1. We again observed reduced cleaved caspase 3 and 7 activity in *Sf3b1* mutant organoids, and a greatly increased survival rate (72% vs 17% surviving organoids, *Figure 3C–E*). Since results from our mouse models indicate that the *Sf3b1$^{K700E}$* mutation already plays a role during early stages of carcinogenesis (*Figure 1G–L*, *Figure 1—figure supplement 1G*), and the tumor-suppressing effect of TGF-β is most prominent on pre-cancerous epithelial cells (*Massagué, 2008*), we additionally established organoid lines with- and without *Sf3b1$^{K700E}$* from non-cancerous mouse pancreata. While *Sf3b1$^{K700E}$* led to slightly reduced proliferation without supplementation of TGF-β1 (*Figure 3—figure supplement*

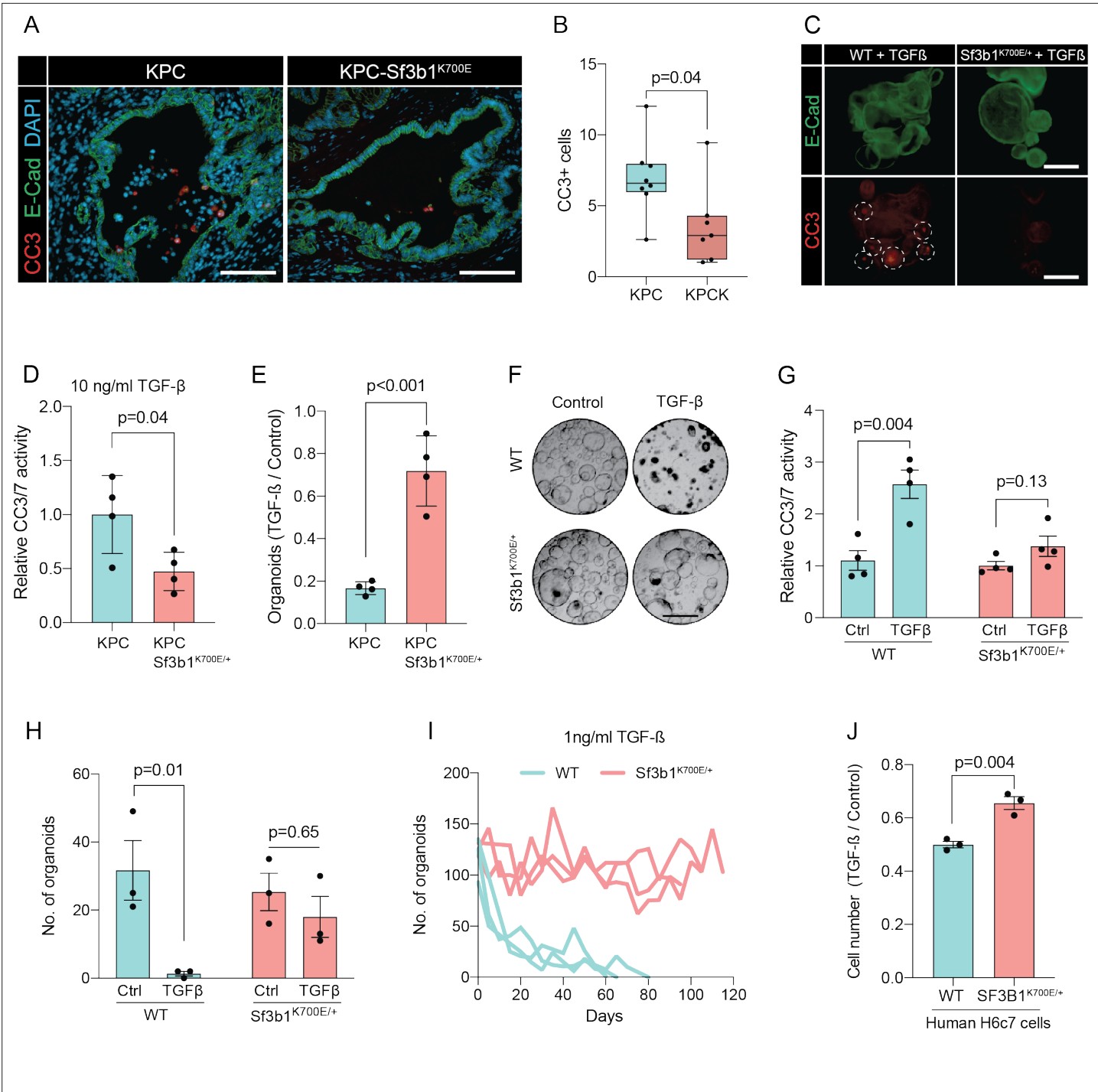

**Figure 3.** *Sf3b1K700E* reduces TGF-β-induced apoptosis. (**A**) Representative microscopy images of E-Cadherin (green) and CC3 (red) in murine PDAC samples. Scale bar is 50 μm. (**B**) Quantification of CC3 positive cells in KPC (n=8) and KPC-Sf3b1K700E/+ (n=7) tumor samples. The average number of CC3 positive cells of 5 microscopic fields is plotted, a two-tailed unpaired t-test was used to compute the indicated p-value. (**C**) Immunofluorescence staining of E-Cadherin (green) and Cleaved Caspase 3 (CC3, red) in WT and Sf3b1K700E/+ organoids exposed to TGF-β1 (10 ng/ml) for 12 hr. CC3-positive cells are highlighted by white dashed lines. Scale bar is 100 μm. (**D**) Quantification of Cleaved Caspase 3 and 7 (CC3/7) activity measured by Caspase-Glo assay of KPC (n=3) and KPC-Sf3b1K700E/+ (n=4) in vitro activated cancer cell lines treated with TGF-β1 (10 ng/ml) for 24 hr. The experiment was repeated independently twice for every cell line, the average of the replicates is shown, error bar represents SEM. Two-tailed unpaired t-test was used to compute the indicated p-value. (**E**) Quantification of viable organoids of the indicated genotype exposed to 10 ng/ml TGF-β1 for 48 hr, normalized to organoid numbers of untreated control samples. Each data point shows a different organoid line. For each organoid line, the experiment was independently performed three times, the average of replicates and SEM is plotted. Two-tailed unpaired t-test was used to compute the indicated

*Figure 3 continued on next page*

*Figure 3 continued*

p-value. (**F**) Representative microscopy images of WT and *Sf3b1*$^{K700E/+}$ organoids exposed to 10 ng/ml TGF-β1 for 48 hr. Scale bar is 500 μM. (**G**) Quantification of CC3/7 in WT and *Sf3b1*$^{K700E/+}$ organoids exposed to 10 ng/ml TGF-β1 for 48 hr. The experiment was repeated independently four times, error bar represents SEM. Two-tailed unpaired t-test was used to compute the indicated p-values. (**H**) Quantification of viable organoids of the indicated genotype exposed to 10 ng/ml TGF-β1 for 48 hr, normalized to organoid numbers of untreated control samples. Each data point shows a different organoid line. For each organoid line, the experiment was independently performed three times, the average of replicates and SEM is plotted. Two-tailed unpaired t-test was used to compute the indicated p-values. (**I**) Organoid count of organoids cultured in medium containing 1 ng/ml TGF-β1 for up to 120 days. One organoid line per genotype was used, the experiment was repeated three times independently. (**J**) Viability of the human pancreatic duct cell line H6c7 overexpressing wildtype or mutated SF3B1 after 72 hr of exposure to 10 ng/ml TGF-β1 assessed by crystal violet staining. The optical density of TGF-β1-treated cells was normalized to untreated control cells. The experiment was independently performed three times, error bar represents SD. A two-tailed unpaired t-test was used to compute the indicated p-value.

The online version of this article includes the following figure supplement(s) for figure 3:

**Figure supplement 1.** *Sf3b1*$^{K700E}$ reduces TGF-β-induced apoptosis.

*1E*), treatment with TGF-β1 led to significantly lower caspase 3 and 7 activity in *Sf3b1*$^{K700E/+}$ vs. wild-type organoids and to significantly enhanced survival (77% vs 3%) (*Figure 3F–H*). Likewise, also in *Kras*$^{G12D/+}$ organoids, SF3B1$^{K700E}$ led to increased survival in the presence of TGF-β1 (*Figure 3—figure supplement 1F*). Finally, exposure to low levels (1 ng/ml and 2 ng/ml) of TGF-β1 also allowed long-term expansion of *Sf3b1*$^{K700E/+}$ organoids (analysed for over 120 days), while the number of *Sf3b1* WT organoids rapidly declined within the first 15 days (*Figure 3I*, *Figure 3—figure supplement 1G, H*).

To assess if reduced sensitivity to TGF-β is also observed in human pancreas cells containing the SF3B1$^{K700E}$ mutation, we stably overexpressed either wildtype or mutated SF3B1 in the human pancreatic duct cell line H6c7. This cell line is derived from healthy pancreatic tissue, which unlike all tested human PDAC-derived cell lines (BxPC-3, Mia PaCa-2, PANC-1, and PSN-1) is still partially responsive to the growth-suppressive effect of TGF-β signalling (*Figure 3J*, *Figure 3—figure supplement 1I*). In line with our results from murine PDAC, overexpression of mutant SF3B1$^{K700E}$ resulted in an increased viability upon TGF-β1 exposure compared to overexpression of wildtype SF3B1 (*Figure 3J*), while no effect of SF3B1$^{K700E}$ on proliferation was observed in absence of TGF-β1 treatment in human pancreatic duct cells and PDAC cell lines (*Figure 3—figure supplement 1J*).

## SF3B1$^{K700E}$ reduces TGF-β sensitivity through *Map3k7* missplicing in organoids and cell lines

To identify how SF3B1$^{K700E}$ could mediate the observed TGF-β resistance in pancreatic cells, we next assessed the impact of the mutation on RNA-splicing. By analysing RNA-seq data from sorted KPC vs. KPC-Sf3b1$^{K700E/+}$ tumor cells we predominantly identified alternative 3' splice events (*Figure 4A*, *Supplementary file 2*), with cryptic 3' ss showing an upstream adenosine enrichment and a less pronounced polypyrimidine tract most frequently located 8–14 bases upstream of the canonical 3' ss (*Figure 4B and C*). These findings are in accordance with previous splice-analyses performed in various murine SF3B1 mutant tissues (*Liu et al., 2021*; *Mupo et al., 2017*; *Obeng et al., 2016*; *Yin et al., 2019*). Next, we sought to determine which of the identified SF3B1$^{K700E}$-dependent alternative splice events are conserved between mice and humans. Due to limited publicly available RNA-seq datasets in human PDAC, we analysed a pan-cancer dataset containing samples of 32 different cancer types. In agreement with previous studies, we found that also in human cancers the SF3B1$^{K700E}$ hotspot mutation leads to a predominant use of cryptic 3' ss (*Figure 4—figure supplement 1A, B*, *Supplementary file 2*; *Alsafadi et al., 2016*; *DeBoever et al., 2015*; *Kesarwani et al., 2017*; *Tang et al., 2020*; *Wang et al., 2016*). Importantly, we further identified 11 genes that contained an alternative 3' splice-event linked to SF3B1$^{K700E}$ in human tumors as well as in KPC mice (*Figure 4D*). Of those genes, *MAP3K7* (formerly known as TGF-β activated kinase 1/*TAK1*) in particular raised our attention. It is a well-described effector of cytokine-signalling that mediates non-canonical TGF-β signalling (*Kim and Choi, 2012*), and it was recently shown to induce EMT and apoptosis in TGF-β-stimulated human mammary cells (*Tripathi et al., 2019*). Using targeted RNA-seq, we discovered that one third of *Map3k7* transcripts were misspliced in pancreata of *Sf3b1*$^{K700E/+}$ and KPC-Sf3b1$^{K700E/+}$ mice (*Figure 4E and F*, *Figure 4—figure supplement 1C*). Confirming inter-species conservation of this alternative splice-event, *MAP3K7* was also misspliced in H6c7 cells and human PDAC cell lines overexpressing SF3B1$^{K700E}$ (*Figure 4F*, *Figure 4—figure supplement 1D*). Notably, when we further assessed

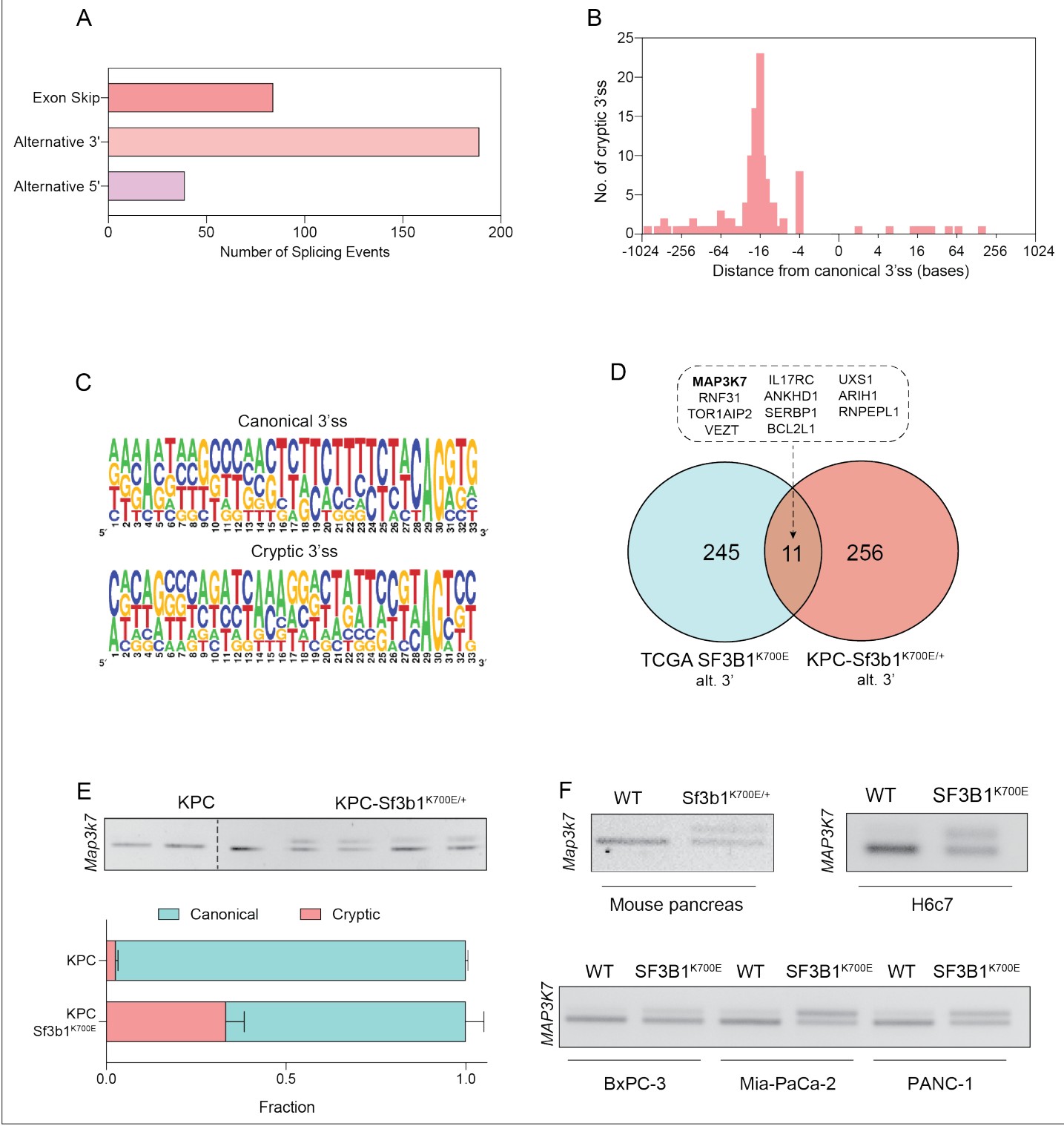

**Figure 4.** SF3B1-K700E induces aberrant splicing of MAP3K7 in human and mouse PDAC. (**A**) Summary of alternative splice events detected in KPC-Sf3b1$^{K700E/+}$ sorted tumor cells (PSI > 0.1, p<0.01). (**B**) Histogram displaying the distance of cryptic 3'splice-site (ss) from the adjacent canonical 3'ss in sorted KPC-Sf3b1$^{K700E/+}$ tumor cells on a logarithmic scale. (**C**) Consensus 3' ss motif in proximity of the canonical (top) and the cryptic (bottom) 3' ss for 7 alternative 3' splicing events identified in sorted KPC-Sf3b1$^{K700E/+}$ tumor cells. (**D**) Venn-diagram depicting alternative 3' splice events in the pan-cancer data-set (PSI > 0.05 and p<1$^{-10}$) and sorted KPC-Sf3b1$^{K700E/+}$ tumor cells (PSI > 0.1, p<0.01). (**E**) Representative gel image (top) and NGS-results (bottom) of *Map3k7* cDNA isolated from sorted KPC (n=3) and KPC-Sf3b1$^{K700E/+}$ tumor cells (n=4). The amplicon includes the 3' splice site of exon 4 and 5, the upper band (114 bp) of the gel image represents the non-canonical transcript variant, the lower band (94 bp) represents the canonical isoform.

*Figure 4 continued on next page*

*Figure 4 continued*

Dashed line indicates cropping of gel image. (**F**) Representative gel image of RT-PCR amplicon of *Map3k7* cDNA isolated from WT and *Sf3b1*$^{K700E/+}$ pancreata and from the indicated human cell lines. The upper band of the gel image represents the non-canonical transcript variant, the lower band represents the canonical isoform. The length of the amplicon derived from murine pancreas cells is 114 bp (canonical) and 137 bp (non-canonical) and from human cell lines 94 bp (canonical) and 114 bp (non-canonical).

The online version of this article includes the following figure supplement(s) for figure 4:

**Figure supplement 1.** SF3B1-K700E induces aberrant splicing of MAP3K7 in human and mouse PDAC.

SF3B1$^{K700E}$ dependent alternative splicing of *Ppp2r5a*, which was reported to impair apoptosis via post-translational modification of BCL2 in leukaemia (*Liu et al., 2020a*), we did not observe significant alternative 3'ss usage or mRNA expression of *Ppp2r5a* in KPC-*Sf3b1*$^{K700E}$ tumor cells (*Figure 4—figure supplement 1E, F*). This observation indicates tissue-specificity of *Ppp2r5a* missplicing.

Since *Sf3b1*$^{K700E}$ dependent missplicing in *MAP3K7* was shown to result in reduced RNA and protein levels of MAP3K7 in leukaemia (*North et al., 2022*), we hypothesized that the reduced responsiveness to TGF-β signalling in SF3B1$^{K700E}$ mutant pancreas cells is caused by lower MAP3K7 levels. Indeed, RT-qPCR and western blotting confirmed a reduction in MAP3K7 levels in vitro and in vivo in *Sf3b1*$^{K700E}$ mutant pancreatic cells (*Figure 5A–D*). Next, we tested whether this reduction could explain the observed resistance to TGF-β in *Sf3b1*-mutant PDAC, and assessed the expression of EMT genes in TGF-β-treated KPC cells with a stable knock-down of MAP3K7 (*Figure 5E*, *Figure 5—figure supplement 1A*). Indeed, a decrease of *Map3k7* mRNA levels to 35% (SD ± 10%) led to a reduced expression in 7 out of 10 EMT genes (*Figure 5—figure supplement 1B*). Moreover, knocking down *Map3k7* in pre-cancerous pancreas organoids led to increased viability upon TGF-β1-treatment (*Figure 5F*, *Figure 5—figure supplement 1C*), and chemical inhibition of p38, one of the major effectors of MAP3K7, partially protected organoids against TGF-β1-induced cell death (*Figure 5G*). Further supporting our hypothesis that *Sf3b1*$^{K700E}$ mediates resistance to TGF-β1 via MAP3K7, overexpression of the full-length isoform of MAP3K7 in TGF-β1-treated *Sf3b1*$^{K700E}$ mutant organoids significantly decreased their viability (*Figure 5H1*).

To further assess conservation of this mechanism between mice and humans, we next treated human pancreatic H6c7 cells as well as human pancreatic organoids before TGF-β1 exposure with Takinib (TAKi), a chemical inhibitor for MAP3K7. In line with our results in murine cells, also in human pancreatic cells inhibition of MAP3K7 led to a significant increase in survival (*Figure 5J and K*). Moreover, conditioning human PANC-1 PDAC cells with TAKi prior to TGF-β1 exposure resulted in a significant decrease in migratory capacity in wound healing assays (*Figure 5L and M*). Taken together, our results suggest that *Sf3b1*$^{K700E}$ mediates resistance of pancreatic epithelial cells to TGF-β1 via MAP3K7, providing a potential mechanism for its role of in PDAC progression.

## Discussion

While the roles of the most frequently mutated tumor driver genes in PDAC are well understood, oncogenes occurring at lower rates are less well studied (*International Cancer Genome Consortium et al., 2010*). One of these genes is SF3B1, for which hotspot mutations occur in various tumor types, including PDAC. While the molecular function of oncogenic SF3B1 on RNA-splicing is well described, how deregulation of misspliced genes contribute to malignancy in different cancer entities is only partially understood. Previous studies have shown that in chronic lymphocytic leukaemia (CLL) SF3B1$^{K700E}$ leads to missplicing of *PPP2R5A*, which in turn stabilizes c-Myc and thereby promotes aggressiveness of tumor cells (*Liu et al., 2020a*; *Yang et al., 2021*). Furthermore, in breast cancer SF3B1$^{K700E}$ causes a tumor-promoting effect through missplicing in *MAP3K7* and downstream activation of NF-κB-signalling (*Liu et al., 2021*).

In our study we analysed the oncogenic function of SF3B1$^{K700E}$ in the context of PDAC. Using a conditional mouse model, we provide the first experimental evidence that *Sf3b1*$^{K700E}$ indeed promotes PDAC progression. While the Sf3b1$^{K700E}$ mutation alone did not induce malignant transformation in the mouse pancreas, co-occurrence with KRAS and p53 mutations increased the severity of PDAC. Effects of SF3B1$^{K700E}$ were already observed during early stages of tumor progression and resulted in reduced expression of TGF-β-responsive EMT genes. Our experiments further revealed that SF3B1$^{K700E}$ reduces TGF-β-induced cell-death in pancreatic duct cell lines and organoids, providing a potential mechanism

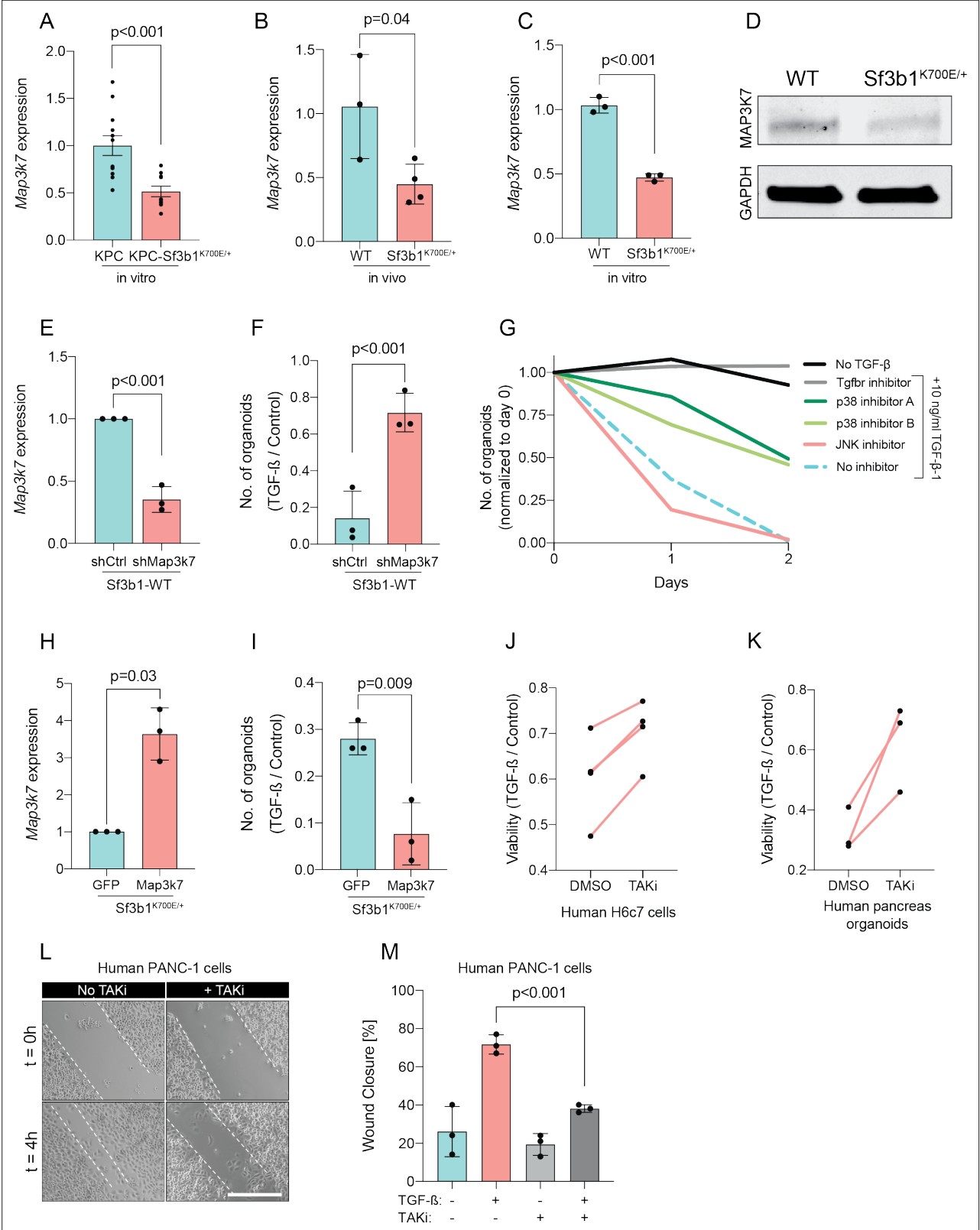

**Figure 5.** Reduction in Map3k7 lowers sensitivity to TGF-β1. (**A–C**) RT-qPCR data showing *Map3k7* expression in KPC (n=13) and KPC-Sf3b1[K700E/+] (n=12) ex vivo tumor cultures (**A**), as well as WT (n=3) and Sf3b1[K700E/+] (n=4) pancreata (**B**) and organoid lines (**C**). For analysis, Ct-values of *Map3k7* were normalized to *Actb* and a two-tailed unpaired t-test was used to compute the indicated p-values. Data show mean and standard error of the mean in A, B and C. (**D**) Representative Western blot gel-image of MAP3K7 (molecular weight 78 kDa) and GAPDH (molecular weight 37 kDa) in WT and Sf3b1[K700E/+]

*Figure 5 continued on next page*

*Figure 5 continued*

organoids. (**E**) RT-qPCR analysis of *Map3k7* in cells transduced with a shMap3k7 compared to a control shRNA. Error bar represents SD, a two-tailed unpaired t-test was used to compute the indicated p-values. (**F**) Quantification of viable WT and *Sf3b1^{K700E/+}* murine pancreatic duct organoids or WT transduced with control shRNA (shCtrl) or shRNA targeting *Map3k7* (shMap3k7). The organoids were exposed to 10 ng/ml TGF-β1 for 24 hr prior to analysis. The experiment was independently performed three times. Error bar represents SD, a two-tailed unpaired t-test was used to compute the indicated p-value. (**G**) Viability of wildtype (WT) organoids cultured in medium containing 10 ng/ml TGF-β1, supplemented with chemical inhibitors targeting the indicated effectors of TGF-β1-signalling. Two independent experiments are summarized. Further details are provided in the methods section. (**H**) RT-qPCR analysis of *Map3k7* in cells transduced by lentivirus with an overexpression construct of *Map3k7*, compared to overexpression of GFP. Error bar represents SD, a two-tailed unpaired t-test was used to compute the indicated p-values. (**I**) Quantification of viable murine pancreatic duct organoids with stable overexpression of *Map3k7*, exposed to 10 ng/ml TGF-β1 for 96 hr. Data represents one organoid line per condition, the experiment was independently performed three times. Error bar represents SD, a two-tailed unpaired t-test was used to compute the indicated p-value. (**J–K**) Viability of human pancreatic duct H6c7 cells (**J**) or human pancreatic duct organoids (**K**) exposed to 10 ng/ml TGF-β1 with addition of the MAP3K7 inhibitor Takinib (TAKi, 5 µM) or DMSO. The viability was assessed after 48 hr of TGF-β1 treatment and normalized to cells grown in absence of TGF-β1. The experiment was independently performed four (**J**) or three (**K**) times. (**L**) Representative micrographs of wound healing assay of PANC-1 cells pre-treated with TGF-β1 (10 ng/ml) for 24 hr, with or without addition of 5 µM TAKi, at the indicated time points after performing the scratch. Scale bar is 50 µM. (**M**) Quantification of wound healing assay shown in (**L**). The experiment was independently performed three times. Error bar represents SD, a two-tailed unpaired t-test was used to compute the indicated p-values.

The online version of this article includes the following figure supplement(s) for figure 5:

**Figure supplement 1.** Reduction in Map3k7 lowers sensitivity to TGF-β1.

for the oncogenicity of SF3B1^{K700E} in PDAC. In line with this hypothesis, pancreatic epithelial cells have previously been found to undergo lethal EMT when stimulated with TGF-β (*David et al., 2016*), and acquiring TGF-β resistance is considered to be essential in early stages of PDAC tumorigenesis (*Hezel et al., 2012*). Interestingly, akin to *Sf3b1^{K700E}*, deletion of the vital TGF-β signalling component *Smad4* only triggered malignant transformation of the pancreas in combination with oncogenic Kras mutations (*Bardeesy et al., 2006b*).

Splicing analysis of RNA-seq datasets from both murine and human cancers identified several genes that are misspliced in the SF3B1^{K700E} background. Among the identified genes was *Map3k7*, which is known to mediate non-canonical TGF-β signaling (*Kim and Choi, 2012*) and to induce EMT and apoptosis in TGF-β stimulated human mammary cells (*Tripathi et al., 2019*). In this study, we show that reducing *Map3k7* levels by shRNA mediated knockdown in pancreatic cell lines and organoids also impairs TGF-β-mediated EMT and cell death, although not to the same extend as *Sf3b1^{K700E}*. Thus, while aberrant splicing of other genes is likely also contributing to the observed resistance to TGF-β, our experiments indicate that TGF-β resistance in *SF3B1^{K700E}* mutant pancreatic ducts is at least partly mediated via MAP3K7. Importantly, this hypothesis is also supported by previous studies, which demonstrated that SF3B1 mutations induce 3' missplicing of *MAP3K7* in various tumor entities (*Bondu et al., 2019*; *Li et al., 2021*; *Lieu et al., 2022*; *Liu et al., 2020b*; *Wang et al., 2016*; *Zhang et al., 2019*), that aberrant splicing by SF3B1^{K700E} reduces MAP3K7 protein levels (*North et al., 2022*), and that MAP3K7 mediates TGF-β-induced EMT and apoptosis in mammary epithelial cells (*Tripathi et al., 2019*). Notably, depending on the cellular context, MAP3K7 has been shown to exert pro-apoptotic stimuli through activation of MAPK p38 and Jun N-terminal kinase (JNK; *Mihaly et al., 2014*), or anti-apoptotic stimuli through activation of the NF-κB pathway (*Mukhopadhyay and Lee, 2020*). Thus, while our results show that SF3B1^{K700E} mediated missplicing and inactivation of MAP3K7 protects pancreatic duct cells from TGF-β induced cell death, in other tissues or in combination with other mutations, the reduction of MAP3K7 levels by SF3B1^{K700E} could trigger different phenotypic responses.

One limitation of our study is the lack of in vivo evidence supporting our hypothesis that SF3B1^{K700E} promotes PDAC progression through MAP3K7 missplicing. We attempted to study the effect of altered MAP3K7 levels in an orthotopic PDAC transplantation model. However, we found that the model was not suitable for adressing this question, since introducing SF3B1^{K700E} in KPC cells had no significant impact on tumor growth after transplantation. This outcome aligned with our expectations, given that KPC cells were isolated from fully established PDAC, where SF3B1^{K700E} does not play the same role as in early-stage PDAC. Additional studies using autochthonous PDAC models, where Map3k7 levels are modulated in early stage KPC or KPC-*Sf3b1^{K700E/+}* tumors, would be required to foster in vivo evidence for the functional role of Map3k7 in SF3B1^{K700E} mutant PDAC.

Taken together, this study provides a first demonstration that oncogenic SF3B1$^{K700E}$ promotes tumor progression in vivo in a mouse model for PDAC. Based on data from pancreatic organoids and cell lines, we further suggest that SF3B1$^{K700E}$ promotes PDAC progression by reducing tumor-suppressive TGF-β signalling through missplicing of MAP3K7.

## Methods

### Animal models

*Sf3b1*$^{K700E/+}$ mice were a gift from E. A. Obeng (Dana-Farber Cancer Institute, Boston, USA) and B. L. Ebert (Brigham and Women's Hospital, Harvard Medical School, Boston, USA). *LSL-Kras*$^{G12D/+}$, *LSL-Trp53*$^{R172H/+}$ and *Ptf1a-Cre* mice were purchased from the Jackson Laboratory (Bar Harbor, Maine, USA). All *Sf3b1*$^{K700E/+}$ and KPC- *Sf3b1*$^{K700E/+}$ mice were bred in a C57BL/6 J background, *Sf3b1*$^{K700E/+}$; *Kras*$^{G12D/+}$ mice were a C57BL/6J-BALB/c strain. Female and male mice were used for all experiments. Animals displaying dwarfism were excluded from analysis. The minimum of animals needed for the study was estimated by Fisher-Yates analysis. Due to the observed variance of the mouse model, more mice than initially estimated were used for the study. For the orthotopic transplantation models, KPC cells were derived from the autochthonous model described above by digestion of fully grown tumors and plating the digest in cell culture dishes. After multiple passages, KPC cells were grown to 80% confluency, and 10$^5$ cells were injected into the pancreas of BALB/cJ mice in matrigel, diluted in a 1:1 ratio with DMEM. 2Two weeks upon transplantation, mice were sacrificed. Mice were held in a specific-pathogen-free (SPF) animal facility at the ETH Phenomics Center EPIC (ETH Zurich, Switzerland). All animal experiments were conducted in accordance with the Swiss Federal Veterinary Office (BVET) guidelines (license no. ZH055/17).

### Cell lines

The human cell lines AsPC-1 (CRL-1682), BxPC-3 (CRL-1687), MIA-PaCa-2 (CRM-CRL-1420), PANC-1 (CRL-1469), PSN-1 (CRL-3211) and HEK293T (CRL-1573) cells were purchased from ATCC. H6c7 cells (ECA001-FP) were purchased from Kerafast. MIA-PaCa-2, PANC-1 and HEK293T cells were maintained in DMEM with 4.5 g/l D-Glucose and GlutaMAX (Gibco), supplemented with 10% fetal calf serum (FCS, Sigma-Aldrich) and 1% Penicillin- streptomycin (P/S, Invitrogen). AsPC-1, BxPC-3, and PSN-1 were cultured in RPMI1640 (Thermo Fisher), supplemented with 10% FCS and 1% P/S. H6c7 cells were cultured in Keratinocyte serum-free medium (Thermo Fisher), supplemented with recombinant EGF and bovine pituitary extract according to the manufacturer's instructions (Thermo Fisher), as well as 100 µg/ml Primocin (Invivogen). Cell lines were regularly checked for mycoplasma-infections by Mycoplasma PCR-detection test (Thermo-Fisher).

### Murine organoids

Murine organoid lines from WT and *Sf3b1*$^{K700E/+}$ animals (B6.129S4) were established as previously described (*Boj et al., 2015*). Briefly, 43-week-old animals of both sexes were euthanized and their pancreata excised. The organs were dissected to thin pieces and digested in 4 mg/ml collagenase IV for 7 min at 37°C. Then, pancreatic ducts were manually picked under a light microscope and seeded in drops of growth factor reduced matrigel. In vitro activated (pre-) cancer organoid lines (*Kras*$^{G12D/+}$, *Kras*$^{G12D/+}$; *Sf3b1*$^{K700E/+}$, *Kras*$^{G12D/+}$; *Trp53*$^{R172H/+}$ and *Kras*$^{G12D/+}$; *Trp53*$^{R172H/+}$; *Sf3b1*$^{K700E/+}$) were established from 8-week-old animals using the same protocol, except that recombination was achieved by delivering Cre-GFP by lentiviral transduction, followed by FACS sorting for GFP positive cells. Tumor-derived KPC and KPC-Sf3b1$^{K700E/+}$ organoid lines were established from solid tumors of KPC or KPC-Sf3b1$^{K700E/+}$ mice. Tumor tissue was digested for 2–3 hr in 4 mg/ml collagenase IV at 37°C, pelleted and seeded in drops of matrigel. The presence of the K700E mutation was validated with Sanger-sequencing on RNA level for each organoid line. Each organoid line was isolated from an individual mouse. Tumor-derived KPC and KPC-Sf3b1$^{K700E/+}$ organoid lines were additionally plated in regular cell culture dishes and grown as monolayer cell culture. Organoids were cultured in organoid medium (OM) composed of AdDMEM/F12 (Gibco) supplemented with GlutaMAX (Gibco), HEPES (Gibco), Penicillin-Streptomycin (Invitrogen), B27 (Gibco), 1.25 mM N-Acetyl-L-cysteine (Sigma), 10 nM Gastrin I (Sigma) and the growth factors: 100 ng/ml FGF10 (Peprotech), 50 ng/ml EGF (Peprotech), 100 ng/

ml Noggin, 100 ng/ml RSPO-1 (Peprotech), and 10 mM Nicotinamide (Sigma). For the first week after duct isolation the culture medium was supplemented with 100 µg/ml Primocin (InvivoGen).

## Immunohistochemistry

Murine tissue specimens were dissected and fixed in 10% neutral buffered formalin for 48–72 hr. Thereafter, formalin was replaced with 70% ethanol before paraffin-embedding and sectioning at a thickness of 4 µm. Hematoxylin and eosin stainings were performed according to the manufacturer's instructions. Anti-Cleaved Caspase 3 (Asp175) antibody (Cell Signaling Technology), anti-Amylase (100–4147) antibody (ThermoFisher), anti-Cytokeratin 19 (ab15463) antibody (ThermoFisher), anti-E-Cadherin (Clone 36) antibody (BD Biosciences), anti-FN-1 antibody (Chemicon), anti-MUC5AC (45M1) antibody (Novus Biologicals) and anti-TNC (Clone 578) antibody was used according to the manufacturer's recommendations. TNC and amylase staining was quantified by calculating the average of Raw Int Density of 3–5 randomly chosen microscopy fields per specimen using ImageJ. CK19 and MUC5AC staining was quantified by counting stained cells of 3–5 randomly chosen microscopy fields per specimen. ADM structures were quantified by counting ADM structures of 5 microscopy fields per specimen of tissue not yet (or only minorly) progressed to PanINs. Luminal necrotic cells were defined as shed cells residing within the lumen of PanIN-lesions.

## Organoid growth assay

Growth of organoids was assessed with CellTiter-Glo 3D (Promega). For absolute quantification of ATP levels, standard curves with defined concentrations of ATP were used for every measurement according to the manufacturer's instructions. As approximation of proliferation rate, the ratio of ATP concentrations at the indicated time points was calculated.

## Crystal violet assay

To measure proliferation of cell lines, 5000 cells were seeded per 96 wells. At the indicated time points, cells were stained with 0.5% (w/v) crystal violet (Sigma-Aldrich) dissolved in an aqueous solution with 20% Methanol (v/v). After washing, plates were allowed to air-dry and the crystal violet was dissolved in 10% acetic acid. Optical density was measured at 595 nM in an Infinite 200 plate reader (Tecan).

## Organoid viability assay

For short-term treatment, organoids were seeded as fragments in 10 µl of Matrigel and allowed to form spheres for 24 hr in regular organoid medium. Organoids were thereafter exposed to the indicated concentration of TGF-β1 (Thermo Fisher). To assess viability of the organoids, intact organoids were counted and compared to untreated organoids after 48 hr of TGF-β1-exposure. This method of quantification was validated by correlating counts of intact organoids with ATP levels as described above (data not shown). For long-term treatment, organoids were seeded as fragments in 40 µl of Matrigel. After allowing to form spheres for 24 hr, the indicated concentration of TGF-β1 was added. After 4 days of TGF-β1-treatment, matrigel-drops were imaged and the number of intact organoids was counted. Then, organoids were reseeded as fragments in normal organoid medium and TGF-β1 was added after 24 hr. Every 4th passage, organoids were split in a 1:1 ratio in the 1 ng/ml TGF-β1 condition.

Commercial cell lines were seeded and TGF-β1 was added at the indicated concentrations 12 hr after plating.

## Organoid invasion assay

Organoids were seeded at equal density in 40 µl of matrigel in 24-well plates. Twenty-four hr after seeding, organoid growth medium was supplemented with 10 ng/ml TGF-β1 (Thermo Fisher). Nienty-six hr after seeding, matrigel domes were detached by rinsing and the migrated organoids (i.e. cells attached to the cell culture dish) were stained by crystal violet. The fraction of attached organoids was calculated by dividing the number of attached organoids by the number of attached organoids plus the number of non-attached organoids (i.e. organoids residing in the matrigel dome).

## Cleaved-caspase 3/7 assay

Organoids were seeded 10 μl of Matrigel and allowed to form spheres for 24 hr in regular organoid medium. Organoids were thereafter exposed to 10 ng/ml TGF-β1 (Thermo Fisher) overnight. Cleavage of Caspase 3 and 7 was quantified by using Caspase-Glo 3/7 Assay System (Promega) according to the manufacturer's instructions.

## Wound healing assay

$8x10^5$ human PANC-1 cells, overexpressing SF3B1-WT or SF3B1$^{K700E}$, were seeded in six-well plates. Ten ng/ml TGF-β1 (and TAKi [5 μM] or DMSO [1:1000] in the respective experiments) was added overnight prior scratch formation with a 10 μl pipette tip. After scratch formation, fresh medium including TGF-β1 (and TAKi or DMSO) was added. 0 hr, 4 hr, and 7 hr after scratch formation, images were taken and the width of the scratch was analysed, performing four measures at standardized positions for each image using ImageJ / Fiji (NIH). To calculate the effect of TGF-β on gap closure in *Figure 2J*, distance of migration (gap width 0 hr after scratch formation – gap width 4 hr after scratch formation) of cells cultured with TGF-β was divided by distance of migration of cells cultured with control medium.

## Chemical inhibitors

The following chemical inhibitors targeting different effectors of the TGF-β-pathway were used: TGFbR-inhibitor A83-01 [50 nM] (Tocris Bioscience), p38-inhibitors SB202190 [10 μM] (Sigma-Aldrich) and SB203580 [10 μM] (Selleckchem), JNK-inhibitor SP600125 [25 μM] (Sigma-Aldrich), SMAD3-inhibitor SIS3 [10 μM] (Sigma-Aldrich) and MAP3K7-inhbitor Takinib [5 μM] (Sigma-Aldrich). The inhibitors were added to the organoid medium directly after seeding.

## shRNA-mediated *Map3k7* knockdown

shRNA targeting murine *Map3k7* was purchased from Sigma-Aldrich (TRCN0000022563). A pLKO.1-puro Non-Target shRNA was used as control. Lentivirus was produced by PEI-based transfection of HEK293T cells. Briefly, HEK293T cells were seeded at 70% confluency in 6-well plates, and the following plasmids were transfected: PAX2 plasmid (1100 ng), VSV-G plasmid (400 ng), cargo plasmid (1500 ng). Medium was changed 12 hr after transfection and the virus-containing supernatant collected after 36 hr. Organoids were dissociated into single cells by Tryp-LE treatment for 5 min at 37°C and consecutive mechanical disruption. After centrifugation, 10% Lentivirus-containing supernatant in organoid medium (v/v) was added to the cell suspension. After a 4–6 hr incubation at 37°C, cells were seeded in matrigel as described above. Organoids were selected in 2 ng/ml Puromycin after the first passage for at least 5 days.

## Overexpression of *Map3k7*

Murine *Map3k7* (full-length isoform) was amplified from cDNA of murine WT duct organoids and cloned into a Lenti-backbone (addgene #73582). Production of lentivirus and transduction of organoids was performed as described above. Organoids stably overexpressing GFP (addgene #17488) were used as experimental control.

## Overexpression of SF3B1-K700E

Codon-optimized human SF3B1-WT and SF3B1-K700E was derived from the plasmids pCDNA3.1-FLAG-SF3B1-WT (addgene #82576) and pCDNA3.1-FLAG-hSF3B1-K700E (addgene #82577) and cloned into a Lenti-backbone. The lentiviruses were produced as described above and used to transduce various cell lines. Puromycin-selection was used to select for transduced cells for at least two passages.

## RNA sequencing

### Cell sorting and RNA extraction

Murine tumors were excised and digested for 2–3 hr in collagenase (4 mg/ml) at 37°C. After addition of fetal calf serum (FCS) to stop the digestion, cells were strained trough a 100 μm and a 70 μm cell strainer. Then, cells were washed twice in PBS + 2% FCS+2 mM EDTA and incubated with mouse FcBlock (BD Biosciences), Epcam-APC (CD326 Monoclonal Antibody (G8.8), APC, eBioscience,

Thermo Fisher) and CD45-BV785 (Clone 30-F11, Biologened) antibodies for 30 min at 4°C. After washing, Epcam-positive-CD45-negative cells were sorted into lysis buffer with a BD FACSAriaIII Cell Sorter (BD Biosciences). Finally, RNA was extracted using NucleoSpin RNA XS kit (Macherey Nagel) according to the manufacturer's instructions.

## Library preparation

The quantity and quality of the isolated RNA was determined with a Qubit (1.0) Fluorometer (Life Technologies, California, USA) and a Tapestion (Agilent, Waldbronn, Germany). The SMARTer Stranded Total RNA-Seq Kit - Pico Input Mammalian (Clontech Laboratories, Inc, A Takara Bio Company,California, USA) was used in the succeeding steps. Briefly, total RNA samples (0.25–10 ng) were reverse-transcribed using random priming into double-stranded cDNA in the presence of a template switch oligo (TSO). When the reverse transcriptase reaches the 5' end of the RNA fragment, the enzyme's terminal transferase activity adds non-templated nucleotides to the 3' end of the cDNA. The TSO pairs with the added non-templated nucleotide, enabling the reverse transcriptase to continue replicating to the end of the oligonucleotide. This results in a cDNA fragment that contains sequences derived from the random priming oligo and the TSO. PCR amplification using primers binding to these sequences can now be performed. The PCR adds full-length Illumina adapters, including the index for multiplexing. Ribosomal cDNA is cleaved by ZapR in the presence of the mammalian-specific R-Probes. Remaining fragments are enriched with a second round of PCR amplification using primers designed to match Illumina adapters.The quality and quantity of the enriched libraries were validated using Qubit (1.0) Fluorometer and the Tapestation (Agilent, Waldbronn, Germany). The product is a smear with an average fragment size of approximately 360 bp. The libraries were normalized to 10 nM in Tris-Cl 10 mM, pH8.5 with 0.1% Tween 20.

## Cluster generation and sequencing

The TruSeq SR Cluster Kit HS4000 or TruSeq PE Cluster Kit HS4000 (Illumina, Inc, California, USA) was used for cluster generation using 8 pM of pooled normalized libraries on the cBOT. Sequencing was performed on the Illumina HiSeq 4000 paired end at 2X126 bp or single end 126 bp using the TruSeq SBS Kit v4-HS (Illumina, Inc, California, USA).

## RNAseq data analysis

Adapters have been trimmed with trimmomatic (v0.35). Pairs for which both reads passed the trimming have been mapped to the murine genome using STAR (v2.7.0a) and indexed BAM files obtained with samtools (v1.9). Reads were counted with featureCounts from subread package (v1.5.0). The read counts have been processed in a statistical analysis using edgeR (v3.24.3), obtaining a list of genes ranked for differential expression by p-value and Benjamini-Hochberg adjusted p-value as the estimate of the false discovery rate. All data is summarized in *Supplementary file 1*.

## Gene set enrichment analysis

Gene set enrichment analysis (v.4.1.0, Broad Institute, MIT) was used to determine enriched gene sets in KPC or KPC-Sf3b1$^{K700E/+}$ tumor cells. Standard parameters of the software were used to perform the analysis. Molecular Signatures Database v7.4, Hallmark Gene Sets (H) was used to query enriched gene sets. The input gene expression matrix contained read-count information (count per million) of 21,633 genes.

## Alternative splicing analysis

We ran a 2-pass alignment of the fastq files using STAR v2.7 (*Dobin et al., 2013*) using the GRCm38. p6 genome as reference. The gene annotation used was GENCODE v.m25. For gene expression quantification, we used a custom script, available at GitHub (copy archived at *Kahles, 2021*); commit hash d074114f1d0a9f518c9cd039f68de0cdf8d583ff. SplAdder v.2.2 (*Kahles et al., 2016*) was run to build splicing graphs and determine splice events. Differential splicing events were determined by calculating a log(psi +x) transformation of the percent spliced in (calculated as ratio of reads supporting the splice event over the number of reads supporting the alternate event). Splice events that did not show any variability over the samples were removed and missing values were mean imputed. After

• Source data 1. Uncropped gel images of Western blot an RT-PCR experiments.

standardization, a two-sided t-test was used to calculate p-values of splice events differences between KPC and KPC-Sf3b1$^{K700E}$ mice. All data is summarized in *Supplementary file 2*.

## Motif analysis
Consensus 3' ss motif in proximity of the canonical and the cryptic 3' ss in sorted KPC-Sf3b1$^{K700E/+}$ tumor cells was assessed by query 30–40 bases spanning the respective 3'ss of the 7 main splice events for a motif using weblogo-sequence creator (https://weblogo.berkeley.edu/logo.cgi).

## RT-PCR and quantitative RT-PCR (qPCR)
RNA-extraction was performed with QIAGEN RNeasy Mini Kit, and cDNA was generated with GoScript Reverse Transcriptase kit (Promega) according to the manufacturers' instructions. RT-PCR was performed with GoTaq G2 Green Master Mix (Promega) and gene-specific primers. Amplicons were fractionated on 2% TBE gel (Life Technologies) supplemented with 0.01% GelRed (Biotium). For qPCR, 2 µL of 1:10-diluted cDNA was added to 8 µl of 5 x HOT FIREPol Evagreen qPCR Supermix (SolisBiodyne). Raw gel images can be found in *Source data 1*. RT-qPCR was performed with a Light-Cycler480 II (Roche). Relative gene expression was determined with the comparative CT method. Genes with a median CT value of more than 33 cycles and a difference of less than 3.3 cycles to the template control (H$_2$O) were defined as not detectable. Sequences of all primers used in this study are listed in *Supplementary file 3*.

## NGS-based isoform quantification of *Map3k7*
Primers generating an amplicon including the exon 4 and exon 5 junction of Map3k7 cDNA were used. Briefly, a gene-specific amplicon was generated in a 20 µL reaction for 35 cycles with GoTaq G2 Green Master Mix (Promega). The PCR product was purified using the NucleoSpin Gel and PCR Clean-up kit (Macherey-Nagel). Thereafter, the isolated product was amplified for eight cycles using primers with sequencing adapters. After column-based isolation of the amplicon and quantification of DNA-yield using a Qubit 3.0 fluorometer and the dsDNA HS assay kit 392 (Thermo Fisher), paired-end sequencing was performed on an Illumina Miseq. The sequencing data was subsequently analyzed with CRISPResso2 (*Clement et al., 2019*).

## Western blotting
Cells were lysed in RIPA buffer, supplemented with Protease Inhibitor (Cell Signaling Technologies) and PhosStop (Sigma-Aldrich) and centrifuged for 10 min at 21,000 *g*. The protein concentration of the supernatant was determined using Pierce BCA assay (ThermoFisher) and a standard curve of albumin. Then, samples were heated for 5' at 95°C in Lämmli buffer and protein lysates were resolved on polyacrylamide Mini-PROTEAN TGX gels (Bio-Rad) and transferred onto nitrocellulose membrane by wet-transfer. The following antibodies were used for immunoblotting: Recombinant anti-GADPH (EPR16891, Abcam) and rabbit monoclonal anti-MAP3K7 (anti-TAK1, D94D7, Cell Signaling Technology) IRDye-conjugated secondary antibodies (donkey anti-goat: LI-COR cat. no. 926–32214; anti-rabbit: LI-COR cat. no. 926–68073) were used for signal detection by an Odyssey Imager (LI-COR) imaging system. Raw gel images can be found in *Source data 1*.

## Acknowledgements
We are grateful to E A Obeng (Dana-Farber Cancer Institute, Boston) and B L Ebert (Brigham and Women's Hospital, Harvard Medical School, Boston) for providing the Sf3b1$^{K700E/+}$ mice used in our study. This work was supported by the Swiss National Science Foundation (grant number 185293 and 176317). The Aceto laboratory is supported by the European Research Council (101001652), the strategic focus area of Personalized Health and Related Technologies at ETH Zurich (PHRT-541), the Swiss National Science Foundation (212183), the Swiss Cancer League (KLS-4834-08-2019), the ETH Lymphoma Challenge (LC-02-22) and the ETH Zürich. TT, AK, KVL are supported by ETH core funding to GR.

# Additional information

## Funding

| Funder | Grant reference number | Author |
|---|---|---|
| Schweizerischer Nationalfonds zur Förderung der Wissenschaftlichen Forschung | 185293 | Gerald Schwank Patrik Simmler |
| Schweizerischer Nationalfonds zur Förderung der Wissenschaftlichen Forschung | 176317 | Markus Stoffel Patrik Simmler |
| European Research Council | 101001652 | Simran Asawa Nicola Aceto |
| ETH Zürich Foundation | PHRT-541 | Simran Asawa Nicola Aceto |
| Schweizerischer Nationalfonds zur Förderung der Wissenschaftlichen Forschung | 212183 | Simran Asawa Nicola Aceto |
| Swiss Cancer League | KLS-4834-08-2019 | Simran Asawa Nicola Aceto |
| ETH Zürich Foundation | LC-02-22 | Simran Asawa Nicola Aceto |
| ETH Zürich Foundation | | Tinu Thomas Andre Kahles Kjong Van-Lehmann Gunnar Rätsch |

The funders had no role in study design, data collection and interpretation, or the decision to submit the work for publication.

## Author contributions

Patrik Simmler, Conceptualization, Data curation, Formal analysis, Investigation, Visualization, Methodology, Writing – original draft; Eleonora I Ioannidi, Data curation; Tamara Mengis, Data curation, Investigation; Kim Fabiano Marquart, Simran Asawa, Cornelia Schwerdel, Investigation; Kjong Van-Lehmann, Andre Kahles, Data curation, Formal analysis, Writing – review and editing; Tinu Thomas, Data curation, Writing – review and editing; Nicola Aceto, Conceptualization, Writing – review and editing; Gunnar Rätsch, Conceptualization, Supervision, Funding acquisition, Writing – review and editing; Markus Stoffel, Conceptualization, Supervision, Writing – review and editing; Gerald Schwank, Conceptualization, Supervision, Writing – original draft, Writing – review and editing

## Author ORCIDs

Patrik Simmler ⓘ http://orcid.org/0000-0003-2883-275X
Markus Stoffel ⓘ http://orcid.org/0000-0003-1304-5817
Gerald Schwank ⓘ http://orcid.org/0000-0003-0767-2953

## Ethics

All animal experiments were conducted in accordance with the Swiss Federal Veterinary Office (BVET) guidelines (license no. ZH055/17).

## Decision letter and Author response

Decision letter https://doi.org/10.7554/eLife.80683.sa1
Author response https://doi.org/10.7554/eLife.80683.sa2

# Additional files

## Supplementary files
- Supplementary file 1. Differential gene expression analysis of sorted KPC and KPC-Sf3b1$^{K700E}$ tumor cells, displaying logFC, logCPM, P-Value, False discovery rate (FDR) and EntrezID.
- Supplementary file 2. Splicing analysis of indicated alternative splice events in sorted KPC and KPC-Sf3b1$^{K700E}$ tumor cells, displaying Ensembl ID, gene name, event positions, percent spliced in (delta psi), standard deviation, p-value and FDR.
- Supplementary file 3. Name and nucleotide sequence of all primers used in the study.
- MDAR checklist

## Data availability
The RNA sequencing raw data of sorted murine cancer cells was deposited in the NCBI Gene Expression Omnibus (GEO) under accession number GSE203339. Splice analysis of human cancers was performed on a previously published dataset, accessible at https://gdc.cancer.gov/about-data/publications/PanCanAtlas-Splicing-2018 (*Kahles et al., 2018*). Material created in this study (i.e. primary cell lines, plasmids) are provided upon request and shall be directed at the corresponding author of this study (Prof. G. Schwank).

The following dataset was generated:

| Author(s) | Year | Dataset title | Dataset URL | Database and Identifier |
|---|---|---|---|---|
| Simmler P | 2022 | RNA-seq of sorted KPC Sf3b1-K700E murine PDAC cells | https://www.ncbi.nlm.nih.gov/geo/query/acc.cgi?acc=GSE203339 | NCBI Gene Expression Omnibus, GSE203339 |

The following previously published dataset was used:

| Author(s) | Year | Dataset title | Dataset URL | Database and Identifier |
|---|---|---|---|---|
| Kahles A | 2018 | Comprehensive Analysis of Alternative Splicing Across Tumors from 8,705 Patients | https://gdc.cancer.gov/about-data/publications/PanCanAtlas-Splicing-2018 | NCI Genomic Data Commons, PanCanAtlas-Splicing-2018 |

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
