## [Editor Report]

This important study investigates the oncogenic and disease promoting potential of the K700E mutation in the splicing factor SF3B1 in a mouse model for pancreatic ductal adenocarcinoma (PDAC), finding that this mutation can promote disease progression both in the presence or absence of Trp53. They further identify SF3B1K700E-induced missplicing of Map3k7 as a critical mechanism that enables isolated pancreas cancer cells to survive TGFβ-induced EMT, senescence and cell death. Together with experiments based on convincing methods that suggest a conserved function in human PDAC, and indicating a specific role for the SF3B1 K700E mutation in early stage tumors, this study makes a valuable contribution to understanding of mechanisms regulating PDAC transformation, and will be interesting to researchers investigating pancreatic cancer.

---

## [Decision Letter]

**Decision letter after peer review:**

Thank you for submitting your article "Mutant SF3B1 promotes PDAC malignancy through TGF-β resistance" for consideration by *eLife*. Your article has been reviewed by 3 peer reviewers, one of whom is a member of our Board of Reviewing Editors, and the evaluation has been overseen by Erica Golemis as the Senior Editor. The reviewers have opted to remain anonymous.

Essential revisions:

1) Additional experimentation testing the validity of findings in vivo is required. All three reviewers requested that data beyond organoids and cell lines be added. For example, orthotopic implantation of the organoids or cells used to examine how the Sf3B1-K700E mutant or Map3k7 expression affects tumor growth and metastasis in vivo is highly recommended as suggested by the reviewers.

2) Increase the human relevance through use of human PDAC lines that have the K700E mutation.

3) Articulate more clearly how the work is novel and distinguishes itself from previous work highlighting Map3k7 as downstream of the Sf3b1-K700E mutation in other cancers.

*Reviewer #1 (Recommendations for the authors):*

Additional analysis of the invasion properties following both expression of Sf3B1-K700E and also following Map3k7 expression, enhanced analysis of the PanINs and PDACs formed in KC and KPC +/- the Sf3b1-K700E mutation, examining effects of Map3k7 on tumors in vivo expressing Sf3b1-K700E would all add to the work significantly.

*Reviewer #2 (Recommendations for the authors):*

Increasing the human relevance of the study would greatly broaden the impact. Some suggestions are listed above. Another possibility would be to seek out cell lines with the SF3BP1 mutation if they are available.

Comment on figures and writing:

A scheme depicting the genotype of the mouse models, specifically of the SF3B1K700E allele would help the reader quickly determine whether the model is a knock-in, a transgenic mouse and so on.

*Reviewer #3 (Recommendations for the authors):*

– The authors propose that SF3B1K700E functions much like SMAD4 mutations in blocking TGFb signaling. But the authors nevertheless propose that SF3B1K700E mutant tumors would nevertheless still benefit from further SMAD4 mutation, even though normally mutations that target the same pathway tend to be mutually exclusive, as SMAD4 may be a more potent block to TGFb signaling. From analysis of patient genomic data, can the authors determine if SF3B1K700E and SMAD4 mutations co-occur or if they are mutually exclusive? If they are mutually exclusive, would this not provide additional compelling patient in vivo evidence that the critical function of SF3B1K700E mutation is to tune TGFb signaling as SMAD4 mutations are no longer occurring in these samples?

– The authors show in figure 2F that KPC-SF3B1K700E cancer cells have a lower expression of EMT genes relative to KPC cell lines and further show in figure 3A that treatment with TGFβ is able to induce upregulation of some of these genes in KPC cells relative to control. As the main premise of the paper in the following figures becomes how the SF3B1K700E mutation can suppress and EMT lethal cell state promoted by TGFβ, it would be useful to see whether TGFβ treatment fails to induce upregulation of these genes in KPC-SF3B1K700E cancer cells relative to untreated control.

– While the authors show that lower expression of MAP3K7 can impair TGFB mediated cell death, a more nuanced discussion of the role MAP3K7 in programmed cell death is warranted, as there is ample literature on this subject, including how MAP3K7 inhibitor Takinib has been used in clinical trials against PDAC (PMID: 31695153, 25146924, 21743023).

– In figure 3B the authors show functional data that suggests SF3B1K700E pancreatic cancer cells have altered cell migration in a TGFβ-dependent context, likely related to the downregulation of EMT related genes observed before. If at all possible, whether these cells also exhibit an altered ability to metastasize in vivo (maybe by gross analysis of metastatic nodules or IHC of micro metastasis in other tissues) should also be commented on when describing the pathology of the mouse model.

– Figure 4 title should be modified to reflect the second main premise of the figure (4D-F), regarding regulation of MAP3K7.

---

## [Author Response]

Essential revisions:1) Additional experimentation testing the validity of findings in vivo is required. All three reviewers requested that data beyond organoids and cell lines be added. For example, orthotopic implantation of the organoids or cells used to examine how the Sf3B1-K700E mutant or Map3k7 expression affects tumor growth and metastasis in vivo is highly recommended as suggested by the reviewers.

We thank the reviewers for raising this important point. To address their concerns, we established and analyzed three in vivo models:

Orthotopic transplantation of tumor-derived KPC and KPC-Sf3b1^K700E^ cells (model of late-stage PDAC tumors)Orthotopic transplantation of KPC cells with shRNA-mediated knockdown of Map3k7Orthotopic transplantation of KPC-Sf3b1^K700E^ cells overexpressing Map3k7

To briefly summarize our results, tumors derived from orthotopically transplanted KPC and KPC-Sf3b1^K700E^ cells did not differ in size (see Figure 1—figure supplement 1M, N, and more detailed explanations in the response to Reviewer #1). Since orthotopic transplantation of KPC cells is a model of late-stage tumors (KPC cells are derived from fully developed PDAC tumors), these results are not surprising and support our initial hypothesis that Sf3b1^K700E^ is more relevant in early stage PDAC rather than late stage PDAC. As expected from these results, also modulating the levels of Map3k7 did not have an effect in tumor growth in orthotopically transplanted KPC or KPC-Sf3b1^K700E^ tumors (Author response image 1).

To further detangle the effect Sf3b1^K700E^ at different tumor stages, we re-analyzed our autochthonous model at an early and late stage during tumor progression. Histological examination at 5 weeks revealed a significant increase in ADM and PanIN incidence (shown by MUC5AC and CK19 IF staining) and a concomitant decrease of acinar cells (shown by b-amylase staining) in KPC-Sf3b1^K700E^ tumors (Figure 1G-J, see Figure 1—figure supplement 1J, K). Analyzing tumors at 9 weeks of age did not show differences in CK19 staining and fibrosis between KPC and KPC-Sf3b1^K700E^ tumors anymore (see Figure 1—figure supplement 1F-I). These findings are in line with our autochthonous KRAS-G12D pre-cancer mouse model, in which Sf3b1^K700E^ leads to a higher incidence of spontaneous progression of PanINs into PDAC (Figure 1L of the revised manuscript).

Taken together, orthotopic transplantation of KPC cells represents a late stage PDAC model, in which Sf3b1^K700E^ does not have an influence on tumor growth anymore. Of note, in a previous study we tried to transplant in vitro induced pancreatic KPC organoids, but failed to observe any tumor growth in vivo. Thus, while using KPC cells isolated from fully established PDAC seems to be a requirement for the orthotopic transplantation, these cells, unfortunately, cannot be used to test whether Sf3b1^K700E^ drives pancreatic tumor progression via MAP3K7. An autochtonous KPC model with downregulated Map3k7 or KPC-Sf3b1^K700E^ model with overexpression of Map3k7 would need to be generated to test this hypothesis in vivo. However, it would take approximately 2 years to generate and breed such models, which would be out of scope for a revision. Therefore, in the revised manuscript we discuss the limitation of our study. In short, we explain that we could only show in vitro in cell lines and organoids that Sf3b1^K700E^ leads to TGF-β resistance in pancreatic duct cells via missplicing of Map3k7, and that it would require further studies to proof that Map3k7 regulation is also relevant for PDAC progression in vivo (line 276-286).

2) Increase the human relevance through use of human PDAC lines that have the K700E mutation.

In the revised manuscript, we present multiple evidence showing the influence of SF3B1^K700E^ on human PDAC lines. To summarize, we found that SF3B1^K700E^ expression leads to Map3k7 missplicing in five human PDAC cell lines and one human pancreatic duct cell line (Figure 4F, see Figure 4—figure supplement 1D). Furthermore, human pancreatic duct cells that overexpress SF3B1^K700E^ demonstrated a higher susceptibility to TGF-β-induced growth restriction compared to cells overexpressing wildtype SF3B1 (Figure 3J). We also executed wound healing assays to empirically evaluate the effect of SF3B1^K700E^ on EMT in human PDAC cells. Consistent with our hypothesis, the results suggest that SF3B1^K700E^ negatively impacts the migratory abilities of PANC-1 cells when treated with TGF-β. We have included these findings in the manuscript (Figure 2I, J). Lastly, we demonstrate that preconditioning human PANC-1 PDAC cells with a Map3k7 inhibitor before exposure to TGF-β1 significantly reduces their migratory capabilities in wound healing assays (Figure 5L, M).

3) Articulate more clearly how the work is novel and distinguishes itself from previous work highlighting Map3k7 as downstream of the Sf3b1-K700E mutation in other cancers.

In the revised manuscript we more clearly state that we are the first to demonstrate that (a) SF3B1^K700E^ increases malignancy and reduces survival in a mouse model of PDAC (line 242-243) and (b) SF3B1^K700E^ decreases sensitivity of murine and human pancreatic duct cells to the tumor suppressive effects of TGF-β (line 246-248). Of note, previous work describing that SF3B1^K700E^ induces missplicing of Map3k7 did not provide a link to TGF-β-resistance and reduced survival. In addition, no other manuscript discussing SF3B1^K700E^ induced missplicing of Map3k7 described the impact of reduced Map3k7 on EMT and apoptosis.

Reviewer #1 (Recommendations for the authors):Additional analysis of the invasion properties following both expression of Sf3B1-K700E and also following Map3k7 expression, enhanced analysis of the PanINs and PDACs formed in KC and KPC +/- the Sf3b1-K700E mutation, examining effects of Map3k7 on tumors in vivo expressing Sf3b1-K700E would all add to the work significantly.

For analyzing the invasion properties, we performed wound healing assays / scratch assays with PANC-1 cells with inducible SF3B1 WT / K700E overexpression. We observed a significant difference in migratory capacity between SF3B1 WT / K700E overexpressing PANC-1 cells when stimulated with TGF-β. We added this data to the revised manuscript (Figure 2I, J). In addition, chemical inhibition of MAP3K7 with Takinib led to a reduced migration capacity in wound healing assays when PANC-1 cells were stimulated with TGF-β (Figure 5L, M).

To enhance the analysis of PanINs and PDAC in early- and late stage KPC and KPC-Sf3b1^K700E^ tumors we assessed expression of CK19, MUC5A1 and b-Amylase by immunofluorescent histochemistry, as well as ADM structures. Significant differences were observed in early-stage tumors but not in late-stage tumors (Figure 1G-J, see Figure 1—figure supplement 1F-K).

To examine the effect of Map3k7 on tumors in vivo, we established orthotopic transplantation models with KPC and KPC-Sf3b1^K700E^ cells, with overexpression and knockdown of Map3k7 (see Author response image 1). However, in contrast to the autochthonous mouse model, orthotopically transplanted KPC and KPC-Sf3b1^K700E^ cells did not show differences in tumor size (see Figure 1—figure supplement 1I, J). These data are not surprising, as orthotopic KPC transplantation is a late-stage cancer model (KPC cells are isolated from fully establish PDAC tumors), and our data suggest that Sf3b1^K700E^ rather plays a role during early stages of PDAC development.

Importantly, our results that Sf3b1^K700E^ does not affect growth of orthotopically transplanted KPC cells means that manipulating Map3k7 levels in this experimental setup should also not affect PDAC growth. Indeed, downregulation of Map3k7 via shRNA in KPC cells yielded tumors of comparable in size to tumors derived from control KPC cells (Author response image 1, B). Moreover, the EMT genes that were differentially expressed in our autochthonous (+/- K700E) mouse models were expressed at similar levels in shMap3k7 KPC tumors vs. control KPC tumors (Author response image 1). Likewise, overexpressing Map3k7 (OE Map3k7) in transplanted KPC-Sf3b1^K700E^ cells led to the development of tumors of comparable size to control tumors without Map3k7 overexpression, again without differential expression of EMT genes between KPC-Sf3b1^K700E^ cells with- and without Map3k7 overexpression (Author response image 1).

**Author response image 1. sa2fig1:** Impact of altered Map3k7 levels in an orthotopic PDAC model. (**A**) Relative gene expression of Map3k7 in KPC cells transduced with shRNA targeting Map3k7 (shMap3k7), normalized to KPC cells transduced with scrambled control shRNA (shCtrl). 3 biological replicates are shown., error bar represents SD. (**B**) Weight of tumors derived by orthotopical transplantation of shMap3k7 and shCtrl KPC cells. 5 biological replicates are shown, error bar represents SEM. (**C**) Relative gene expression of EMT genes in tumors derived by orthotopic transplantation of shCtrl and shMap3k7 cells. 4 biological replicates are shown. (**D**) Relative gene expression of Map3k7 in KPC-Sf3b1^K700E^ cells transduced with an overexpression vector of Map3k7 (OE Map3k7), normalized to control KPC cells without Map3k7 overexpression. 3 biological replicates are shown, error bar represents SD. A two-sided student’s t-test was used to calculate significance. (**E**) Weight of tumors derived by orthotopical transplantation of Map3k7 overexpressing KPC-Sf3b1^K700E^ cells (n=5) and control KPC-Sf3b1^K700E^ cells (n=4). Error bar represents SEM. (**F**) Relative gene expression of EMT genes in tumors derived by orthotopic transplantation of KPC-Sf3b1^K700E^ cells with- and without overexpression of Map3k7. 4 biological replicates are shown. A two-sided student’s t-test was used to calculate significance in Figure 2A-F.

Reviewer #2 (Recommendations for the authors):Increasing the human relevance of the study would greatly broaden the impact. Some suggestions are listed above. Another possibility would be to seek out cell lines with the SF3BP1 mutation if they are available.

In the revised manuscript we present data on missplicing of MAP3K7 in 5 human PDAC cell lines and one human pancreatic duct cell line stably overexpressing SF3B1-K700E (Figure 4F). We further demonstrate that the human pancreatic duct cell line overexpressing SF3B1-K700E is less sensitive to growth inhibition by TGF-β than SF3B1-WT overexpressing cells, while the PDAC cell lines are insensitive to TGF-β, regardless of their genotype (Figure 3J, see Figure 3—figure supplement I). These data support our hypothesis that SF3B1-K700E decreases sensitivity to TGF-β induced apoptosis/growth inhibition in pancreatic duct cells, but that in late stage PDAC misregulation of other TGF-β modulating factors plays a more important role.

Furthermore, performing wound healing assays / scratch assays with human PANC-1 cells with inducible SF3B1-WT/ K700E overexpression revealed a significant difference in migratory capacity when stimulated with TGF-β. We added this data to the revised manuscript (Figure 2I, J).

Comment on figures and writing:A scheme depicting the genotype of the mouse models, specifically of the SF3B1K700E allele would help the reader quickly determine whether the model is a knock-in, a transgenic mouse and so on.

We added a schematic overview of the SF3B1^K700E^ knock-in to the revised manuscript (Figure 1A).

Reviewer #3 (Recommendations for the authors):– The authors propose that SF3B1K700E functions much like SMAD4 mutations in blocking TGFb signaling. But the authors nevertheless propose that SF3B1K700E mutant tumors would nevertheless still benefit from further SMAD4 mutation, even though normally mutations that target the same pathway tend to be mutually exclusive, as SMAD4 may be a more potent block to TGFb signaling. From analysis of patient genomic data, can the authors determine if SF3B1K700E and SMAD4 mutations co-occur or if they are mutually exclusive? If they are mutually exclusive, would this not provide additional compelling patient in vivo evidence that the critical function of SF3B1K700E mutation is to tune TGFb signaling as SMAD4 mutations are no longer occurring in these samples?

We performed a mutual exclusivity analysis with SF3B1-K700E and components of the TGF-β pathway in published human PDAC databases (www.cbioportal.org), but did not observe mutual exclusivity between these genes. Of note, the SF3B1-K700E mutation is rare, and due to the limited sample size (only 7 samples contain the SF3B1-K700E mutation) the value of the analysis is limited.

**Author response table 1. sa2table1:** Mutual exclusivity analysis of public PDAC databases (ICGC, CPTAC, QCMG, TCGA, UTSW), including 910 patients. Mutation frequency is 25% for SMAD4, 5% for TGF-ΒR2, 3% for SMAD2, 2.6% for TGF-ΒR1, 1.4% for SMAD3, 0.7% for SF3B1-K700E, 0.7% for TGF-ΒR3, 0.4% for SMAD1. Analysis was performed on cbioportal.org.

A	B	p-Value	q-Value	Tendency
SF3B1: MUT=K700E	SMAD1	0.974	0.974	Mutual exclusivity
SF3B1: MUT=K700E	SMAD2	0.823	0.974	Mutual exclusivity
SF3B1: MUT=K700E	SMAD3	0.917	0.974	Mutual exclusivity
SF3B1: MUT=K700E	SMAD4	0.524	0.974	Mutual exclusivity
SF3B1: MUT=K700E	TGF-ΒR1	0.851	0.974	Mutual exclusivity
SF3B1: MUT=K700E	TGF-ΒR2	0.268	0.682	Co-occurrence
SF3B1: MUT=K700E	TGF-ΒR3	0.961	0.974	Mutual exclusivity

– The authors show in figure 2F that KPC-SF3B1K700E cancer cells have a lower expression of EMT genes relative to KPC cell lines and further show in figure 3A that treatment with TGFβ is able to induce upregulation of some of these genes in KPC cells relative to control. As the main premise of the paper in the following figures becomes how the SF3B1K700E mutation can suppress and EMT lethal cell state promoted by TGFβ, it would be useful to see whether TGFβ treatment fails to induce upregulation of these genes in KPC-SF3B1K700E cancer cells relative to untreated control.

For the revised manuscript, we assessed the expression of the EMT-gene set of KPC and KPC-SF3B1^K700E^ cells stimulated with TGF-β. Indeed, 6 of 9 genes showed no or impaired upregulation in KPC-SF3B1^K700E^ cells in comparison to KPC-SF3B1 (see Figure 2—figure supplement 1G)

– While the authors show that lower expression of MAP3K7 can impair TGFB mediated cell death, a more nuanced discussion of the role MAP3K7 in programmed cell death is warranted, as there is ample literature on this subject, including how MAP3K7 inhibitor Takinib has been used in clinical trials against PDAC (PMID: 31695153, 25146924, 21743023).

In the revised manuscript, we discuss the multi-faceted role of MAP3K7 in apoptosis (line 269-275).

– In figure 3B the authors show functional data that suggests SF3B1K700E pancreatic cancer cells have altered cell migration in a TGFβ-dependent context, likely related to the downregulation of EMT related genes observed before. If at all possible, whether these cells also exhibit an altered ability to metastasize in vivo (maybe by gross analysis of metastatic nodules or IHC of micrometastasis in other tissues) should also be commented on when describing the pathology of the mouse model.

We analyzed macroscopic metastatic nodules in the liver of KPC and KPC-SF3B1^K700E^ mice. We found that 2 out of 32 KPC mice vs. 5 of 22 KPC-SF3B1^K700E^ mice developed macro-metastases in liver (of animals older than 56 days). However, this difference is statistically not significant (chi-square test, p=0.37), and the role of EMT in metastasis formation is complex and controversial. We therefore did not add this data to the manuscript.

– Figure 4 title should be modified to reflect the second main premise of the figure (4D-F), regarding regulation of MAP3K7.

We changed the title of Figure 4 accordingly (line 806).